# ON PROXIMAL POLICY OPTIMIZATION'S HEAVY-TAILED GRADIENTS

## ABSTRACT

Modern policy gradient algorithms, notably Proximal Policy Optimization (PPO), rely on an arsenal of heuristics, including loss clipping and gradient clipping, to ensure successful learning. These heuristics are reminiscent of techniques from robust statistics, commonly used for estimation in outlier-rich ("heavy-tailed") regimes. In this paper, we present a detailed empirical study to characterize the heavy-tailed nature of the gradients of the PPO surrogate reward function. We demonstrate pronounced heavy-tailedness of the gradients, specifically for the *actor* network, which increases as the current policy diverges from the behavioral one (i.e., as the agent goes further off policy). Further examination implicates the likelihood ratios and advantages in the surrogate reward as the main sources of the observed heavy-tailedness. Subsequently, we study the effects of the standard PPO *clipping heuristics*, demonstrating how these tricks primarily serve to offset heavy-tailedness in gradients. Motivated by these connections, we propose incorporating GMOM, a high-dimensional robust estimator, into PPO as a substitute for three clipping tricks. Our method achieves comparable performance to that of PPO with all heuristics enabled on a battery of MuJoCo continuous control tasks.

## 1 INTRODUCTION

As Deep Reinforcement Learning (DRL) methods have made strides on such diverse tasks as game playing and continuous control (Berner et al., 2019; Silver et al., 2017; Mnih et al., 2015), policy gradient methods (Williams, 1992; Sutton et al., 2000; Mnih et al., 2016) have emerged as a popular alternative to dynamic programming approaches. Since the breakthrough results of Mnih et al. (2016) demonstrated the applicability of policy gradients in DRL, a number of popular variants have emerged (Schulman et al., 2017; Espeholt et al., 2018). Proximal Policy Optimization (PPO) (Schulman et al., 2017)—one of the most popular policy gradient methods—introduced the clipped importance sampling update, an effective heuristic for off-policy learning. However, while their stated motivation for clipping draws upon trust-region enforcement, the behavior of these methods tends to deviate from its key algorithmic principle (Ilyas et al., 2018), and exhibit sensitivity to implementation details (Engstrom et al., 2019). More generally, policy gradient methods are brittle, sensitive to both the random seed and hyperparameter choices, and poorly understood (Ilyas et al., 2018; Engstrom et al., 2019; Henderson et al., 2017; 2018; Islam et al., 2017). The ubiquity of these issues raises a broader concern about our understanding of policy gradient methods.

In this work, we take a step forward towards understanding the workings of PPO, the most prominent and widely used deep policy gradient method. Noting that the heuristics implemented in PPO are evocative of estimation techniques from robust statistics in *outlier-rich* settings, we conjecture that the heavy-tailed distribution of gradients is the main obstacle addressed by these heuristics. We perform a rigorous empirical study to understand the causes of heavy-tailedness in PPO gradients. Furthermore, we provide a novel perspective on the clipping heuristics implemented in PPO by showing that these heuristics primarily serve to alleviate heavy-tailedness in gradients.

Our first contribution is to analyze the role played by each component of the PPO objective in the heavy-tailedness of the gradients. We observe that as the training proceeds, gradients of both the actor and the critic loss get more heavy-tailed. Our findings show that during *on-policy* gradient steps the advantage estimates are the primary contributors to the heavy-tailed nature of the gradients. Moreover, as *off-policyness* increases (i.e. as the behavioral policy and actor policy diverge) dur-

ing training, the likelihood-ratios that appear in the surrogate objective exacerbates the heavy-tailed behavior. Subsequently, we demonstrate that the clipping heuristics present in standard PPO implementations (i.e., gradient clipping, actor objective clipping, and value loss clipping) significantly counteract the heavy-tailedness induced by off-policy training. Finally, motivated by this analysis, we present an algorithm that uses Geometric Median-of-Means (GMOM), a high-dimensional robust aggregation method adapted from the statistics literature. Without using any of the objective clipping and gradient clipping heuristics implemented in PPO, the GMOM algorithm nearly matches PPO's performance on MuJoCo (Todorov et al., 2012) continuous control tasks.

## 2 PRELIMINARIES

We define a Markov Decision Process (MDP) as a tuple $(\mathcal{S}, \mathcal{A}, R, \gamma, P)$, where $\mathcal{S}$ represent the set of environments states, $\mathcal{A}$ represent the set of agent actions, $R : \mathcal{S} \times \mathcal{A} \to \mathbb{R}$ is the reward function, $\gamma$ is the discount factor, and $P : \mathcal{S} \times \mathcal{A} \times \mathcal{S} \to \mathbb{R}$ is the state transition probability distribution. The goal in reinforcement learning is to learn a policy $\pi_\theta : \mathcal{S} \times \mathcal{A} \to \mathbb{R}_+$, parameterized by $\theta$, such that the expected cumulative discounted reward (known as returns) is maximized. Formally, $\pi^* := \operatorname{argmax}_\pi \mathbb{E}_{a_t \sim \pi(\cdot|s_t), s_{t+1} \sim P(\cdot|s_t, a_t)} \left[ \sum_{t=0}^{\infty} \gamma^t R(s_t, a_t) \right]$.

Policy gradient methods directly parameterize the policy (also known as *actor* network). Since directly optimizing the cumulative rewards can be challenging, modern policy gradient algorithms typically optimize a surrogate reward function. Often the surrogate objective includes a likelihood ratio to allow importance sampling from a behavior policy $\pi_0$ while optimizing policy $\pi_\theta$. For example, Schulman et al. (2015a) optimize:

$$\max_\theta \mathbb{E}_{(s_t, a_t) \sim \pi_0} \left[ \frac{\pi_\theta(a_t, s_t)}{\pi_0(a_t, s_t)} A_{\pi_0}(s_t, a_t) \right] , \tag{1}$$

where $A_{\pi_\theta} = Q_{\pi_\theta}(s_t, a_t) - V_{\pi_\theta}(s_t)$. Here, Q-function , i.e. $Q_{\pi_\theta}(s, a)$, is the expected discounted reward after taking an action $a$ at state $s$ and following $\pi_\theta$ afterwards and $V_{\pi_\theta}(s)$ is the value estimate (implemented with a *critic* network).

However, the surrogate is indicative of the true reward function only when $\pi_\theta$ and $\pi_0$ are close in distribution. Different policy gradient methods (Schulman et al., 2015a; 2017; Kakade, 2002) attempt to enforce the closeness in different ways. In Natural Policy Gradients (Kakade, 2002) and Trust Region Policy Optimization (TRPO) (Schulman et al., 2015a), authors utilize a conservation policy iteration with an explicit divergence constraint which provides provable lower bounds guarantee on the improvements of the parameterized policy. On the other hand, PPO (Schulman et al., 2017) implements a clipping heuristic on the likelihood ratio of the surrogate reward function to avoid excessively large policy updates. Specifically, PPO optimizes the following objective:

$$\max_\theta \mathbb{E}_{(s_t, a_t) \sim \pi_0} \left[ \min \left( \text{clip}(\rho_t, 1 - \epsilon, 1 + \epsilon) \hat{A}_{\pi_0}(s_t, a_t), \rho_t \hat{A}_{\pi_0}(s_t, a_t) \right) \right] , \tag{2}$$

where $\rho_t := \frac{\pi_\theta(a_t, s_t)}{\pi_0(a_t, s_t)}$. We refer to $\rho_t$ as *likelihood-ratios*. Due to a minimum with the unclipped surrogate reward, the PPO objective acts as a pessimistic bound on the true surrogate reward. As in standard PPO implementation, we use Generalized Advantage Estimation (GAE) (Schulman et al., 2015b). Moreover, instead of fitting the value network via regression to target values:

$$L^V = (V_{\theta_t} - V_{targ})^2, \tag{3}$$

standard implementations fit the value network with a PPO-like objective:

$$L^V = \max \left\{ (V_{\theta_t} - V_{targ})^2, \left( \text{clip} \left( V_{\theta_t}, V_{\theta_{t-1}} - \varepsilon, V_{\theta_{t-1}} + \varepsilon \right) - V_{targ} \right)^2 \right\}, , \tag{4}$$

where $\epsilon$ is the same value used to clip probability raitos in PPO's loss function (Eq. 9).

PPO uses the following training procedure: At any iteration $t$, the agent creates a clone of the current policy $\pi_{\theta_t}$ which interacts with the environment to collect rollouts $\mathcal{B}$ (i.e., state-action pairs $\{(s_i, a_i)\}_{i=1}^N$). Then the algorithm optimizes the policy $\pi_\theta$ and value function $V_\theta$ for a fixed $K$ gradient steps on the sampled data $\mathcal{B}$. Since at every iteration the first gradient step is taken on the same policy from which the data was sampled, we refer to these gradient updates as *on-policy* steps.

And as for the remaining $K - 1$ steps, the sampling policy differs from the current agent, we refer to these updates as *off-policy* steps.

Throughout the paper, we consider a stripped-down variant of PPO (denoted PPO-NoCLIP) that consists of policy gradient with importance weighting (Eq. 1), but has been simplified as follows: i) no likelihood-ratio clipping, i.e., no *objective function clipping*; ii) value network optimized via regression to target values (Eq. 3) without *value function clipping*; and iii) no *gradient clipping*. Overall PPO-NoCLIP uses the objective summarized in App. A. One may argue that since PPO-NoCLIP removes the clipping heuristic from PPO, the unconstrained maximization of Eq. 1 may lead to excessively large policy updates. In App. I, we empirically justify the use of Eq. 1 by showing that with the small learning rate used in our experiments (tuned hyperparameters in Table 1), PPO-NoCLIP maintains a KL based trust-region like PPO throughout the training. We elaborate this in App. I.

## 2.1 FRAMEWORK FOR ESTIMATING HEAVY-TAILEDNESS

We now formalize our setup for studying the distribution of gradients. Throughout the paper, we use the following definition of the heavy-tailed property:

**Definition 1** (Resnick (2007)). *A non-negative random variable $w$ is called* heavy-tailed *if its tail probability $F_w(t) := P(w \geq t)$ is asymptotically equivalent to $t^{-\alpha^*}$ as $t \to \infty$ for some positive number $\alpha^*$. Here $\alpha^*$ determines the heavy-tailedness and $\alpha^*$ is called tail index of $w$.*

For a heavy-tailed distribution with index $\alpha^*$, its $\alpha$-th moment exists only if $\alpha < \alpha^*$, i.e., $\mathbb{E}[w^\alpha] < \infty$ iff $\alpha < \alpha^*$. A value of $\alpha^* = 1.0$ corresponds to a Cauchy distribution and $\alpha^* = \infty$ (i.e., all moments exist) corresponds to a Gaussian distribution. Intuitively, as $\alpha^*$ decreases, the central peak of the distribution gets higher, the valley before the central peak gets deeper, and the tails get heavier. In other words, the lower the tail-index, the more heavy-tailed the distribution. However, in the finite sample setting, estimating the tail index is notoriously challenging (Simsekli et al., 2019; Danielsson et al., 2016; Hill, 1975).

In this study, we explore three estimators as heuristic measures to understand heavy tails and non-Gaussianity of gradients (refer to App. B for details). *(i) Alpha-index estimator* which measures alpha-index for symmeteric $\alpha$-stable distributions. This estimator is derived under the (strong) assumption that the stochastic Gradient Noise (GN) vectors are coordinate-wise independent and follow a symmetric alpha-stable distribution. *(ii) Anderson-Darling test* (Anderson & Darling, 1954) on random projections of GN to perform Gaussianity testing (Panigrahi et al., 2019). To our knowledge, the deep learning literature has only explored these two estimators for analyzing the heavy-tailed nature of gradients. *(iii)* Finally, in our work, we propose using *Kurtosis*. To quantify the heavy-tailedness relative to a normal distribution, we measure kurtosis (fourth standardized moment) of the gradient norms. Given samples $\{X_i\}_{i=1}^N$, the kurtosis $\kappa$ is given by:

$$\kappa = \frac{\sum_{i=1}^N (X_i - \bar{X})^4 / N}{\left( \sum_{i=1}^N (X_i - \bar{X})^2 / N \right)^2},$$

where $\bar{X}$ is the empirical mean of the samples. With a slight breach of notation, we use kurtosis to denote $\kappa^{1/4}$. It is well known that for a Pareto distribution with shape $\alpha \geq 4$, the lower the tail-index (shape parameter $\alpha$) the higher the kurtosis. For $\alpha < 4$, since the fourth moment is non-existent, kurtosis is infinity. While for Gaussian distribution, the kurtosis value is approximately $1.31$. In App. B, we show behavior of kurtosis on Gaussian and Pareto data with varying sample sizes and tail-indices for Pareto data.

## 3 HEAVY-TAILEDNESS IN POLICY-GRADIENTS: A CASE STUDY ON PPO

We now examine the distribution of gradients in PPO. To start, we examine the behavior of gradients at only on-policy steps. We fix the policy at the beginning of every training iteration and just consider the gradients for the first step (see App. D for details). As the training proceeds, the gradients clearly become more heavy-tailed (Fig. 1(a)). To thoroughly understand this behavior and the contributing factors, we separately analyze the contributions from different components in the loss function. We also separate out the contributions coming from actor and critic networks.

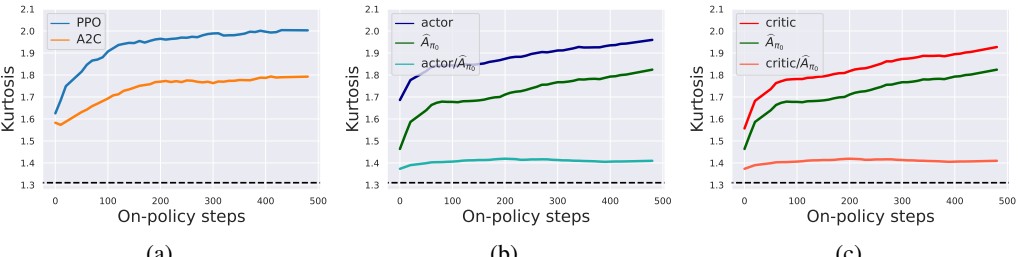

Figure 1: **Heavy-tailedness in PPO during on-policy iterations**. All plots show mean kurtosis aggregated over 8 MuJoCo environments. For other estimators, see App. F. For individual environments with error bars, see App. H. Increases in Kurtosis implies an increase in heavy-tailedness. Dotted line represents the Kurtosis value for a Gaussian distribution. (a) Kurtosis vs on-policy iterations for A2C and PPO. Evidently, as training proceeds, the gradients become more heavy-tailed for both the methods. (b) Kurtosis vs on-policy iterations for actor networks in PPO. (c) Kurtosis vs on-policy iterations for critic networks in PPO. Both critic and actor gradients become more heavy-tailed as the agent is trained. Note that as the gradients become more heavy-tailed, we observe a corresponding increase of heavy-tailedness in the advantage estimates ($\hat{A}_{\pi_0}$). However, "actor/$\hat{A}_{\pi_0}$" and "critic/$\hat{A}_{\pi_0}$" (i.e., actor or critic gradient norm divided by advantage) remain light-tailed throughout the training. In App. E, we perform ablation tests to highlight the reason for heavy-tailed behavior of advantages.

Alongside, we also perform ablations to understand how PPO heuristics affect the heavy-tailed nature of the gradient distribution. To decouple the behavior of naïve policy gradients from PPO optimizations, we consider a variant of PPO which we call PPO-NOCLIP as described in Section 2. Recall that in a nutshell PPO-NOCLIP implements policy gradient with just importance sampling. In what follows, we perform a fine-grained analysis of PPO at on-policy iterations.

### 3.1 HEAVY-TAILEDNESS IN ON-POLICY TRAINING

Given the trend of increasing heavy-tailedness in on-policy gradients, we first separately analyze the contributions of the actor and critic networks. On both these component network gradients, we observe similar trends, with the heavy-tailedness in the actor gradients being marginally higher than the critic network (Fig. 1). Note that during on-policy steps, since the likelihood-ratios are just 1, the gradient of actor network is given by $\nabla_\theta \log\left(\pi_\theta(a_t, s_t)\right) \hat{A}_{\pi_0}(s_t, a_t)$ and the gradient of the critic network is given by $\nabla_\theta V_\theta \hat{A}_{\pi_0}(s_t, a_t)$ where $\pi_0$ is the behavioral policy. To explain the rising heavy-tailed behavior, we separately plot the advantages $\hat{A}_{\pi_0}$ and the advantage divided gradients ( i.e, $\nabla \log(\pi_\theta(a_t|s_t))$ and $\nabla_\theta V_\theta$). Strikingly, we observe that while the advantage divided gradients are not heavy-tailed for both value and policy network, the heavy-tailedness in advantage estimates increases as training proceeds. This elucidates that during on-policy updates, outliers in advantage estimates are the only source of heavy-tailedness in actor and critic networks.

To understand the reasons behind the observed characteristic of advantages, we plot value estimates as computed by the critic network and the discounted returns used to calculate advantages (Fig. 7 in App. E) We don't observe any discernable heavy-tailedness trends in value estimates and a slight increase in returns. However, remarkably, we notice a very similar course of an increase in heavy-tailedness with negative advantages (whereas positive advantages remained light-tailed) as training proceeds. In App. E.3, we also provide evidence to this observation by showing the trends of increasing heavy-tailed behavior with the histograms of $\log(|A_{\pi_\theta}|)$ grouped by their sign as training proceeds for one MuJoCo environment (HalfCheetah-v2). This observation highlights that, at least in MuJoCo continuous control environments, there is a positive bias of the learned value estimate for actions with negative advantages. And in addition, our experiments also suggest that the outliers in advantages (primarily, in negative advantages) are the root cause of observed heavy-tailed behavior in the actor and critic gradients.

We also analyse the gradients of A2C (Mnih et al., 2016)—an on-policy RL algorithm—and observe similar trends (Fig. 1(a)), but at a relatively smaller degree of heavy-tailedness. Although they start

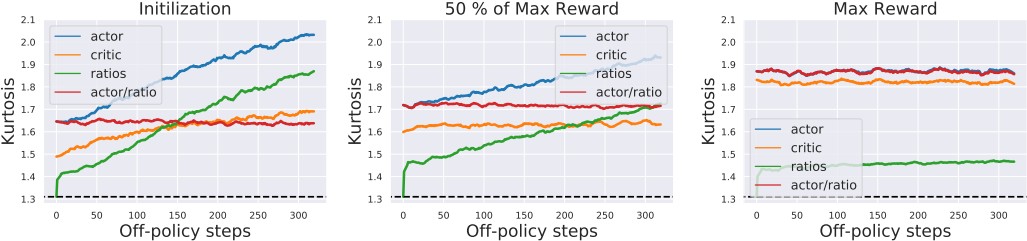

Figure 2: **Heavy-tailedness in PPO-NoClip during off-policy steps** at various stages of training iterations in MuJoCo environments. All plots show mean kurtosis aggregated over 8 Mujoco environments. Plots for other estimators can be found in App. F. We also show trends with these estimators (with error bars) on individual environments in App H. Increases in Kurtosis implies an increase in heavy-tailedness. Dotted line represents the Kurtosis value for a Gaussian distribution. Note that the analysis is done with gradients taken on a fixed batch of data within a single iteration. As off-policyness increases, the actor gradients get substantially heavy-tailed. This trend is corroborated by the increase of heavy-tailedness in ratios. Moreover, consistently we observe that the heavy-tailedness in "actor/ratios" stays constant. While initially during training, the heavy-tailedness in the ratio's increases substantially, during later stages the increase tapers off. The overall increase across training iterations is explained by the induced heavy-tailedness in the advantage estimates (cf. Sec. 3.1).

at a similar magnitude, the heavy-tailed nature escalates at a higher rate in PPO[1]. This observation may lead us to ask: What is the cause of heightened heavy-tailedness in PPO (when compared with A2C)? Next, we demonstrate that off-policy training can exacerbate the heavy-tailed behavior.

### 3.2 OFFPOLICYNESS ESCALTE HEAVYTAILNESS IN GRADIENTS

To analyze the gradients at off-policy steps, we perform the following experiment: At various stages of training (i.e., at initialization, 50% of maximum reward, and maximum reward), we fix the actor and the critic network at each gradient step during off-policy training and analyze the collected gradients (see App. D for details). First, in the early stages of training, as the off-policyness increases, the heavy-tailedness in gradients (both actor and critic) increases. However, unlike with on-policy steps, actor gradients are the major contributing factor to the overall heavy-tailedness of the gradient distribution. In other words, the increase in heavy-tailedness for actor gradients due to off-policy training is substantially greater than for critic gradients (Fig. 2). Furthermore, this increase lessens in later stages of training as the agent approaches its maximum performance.

Now we turn our attention to explaining the possible causes for such a profound increase. The strong increase in heavy-tailedness of the actor gradients during off-policy training coincides with a increase of heavy-tailedness in the distribution of likelihood ratios $\rho$, given by $\pi_\theta(a_t, s_t)/\pi_0(a_t, s_t)$. The corresponding increase in heavy-tailedness in ratios can be explained theoretically. In continuous control RL tasks, the actor-network often implements the policy with a Gaussian distribution where the policy parameters estimate the mean and the (diagonal) covariance. With a simple example, we highlight the heavy-tailed behavior of such likelihood-ratios of Gaussian density function. This example highlights how even a minor increase in the standard deviation of the distribution of the current policy (as compared to behavior policy) can induce heavy-tails.

**Example 1** (Wang et al., 2018). Assume $\pi_1(x) = \mathcal{N}\left(x; 0, \sigma_1^2\right)$ and $\pi_2(x) = \mathcal{N}\left(x; 0, \sigma_2^2\right)$. Let $\rho = \pi_1(x)/\pi_2(x)$ at a sample $x \sim \pi_2$. If $\sigma_1 \leq \sigma_2$, then likelihood ratio $\rho$ is bounded and its distribution is not heavy-tailed. However, when $\sigma_1 > \sigma_2$, then $w$ has a heavy-tailed distribution with the tail-index (Definition 1) $\alpha^* = \sigma_1^2/(\sigma_1^2 - \sigma_2^2)$.

---

[1]In Appendix E.2, we show a corresponding trend in the heavy-tailedness of advantage estimates.

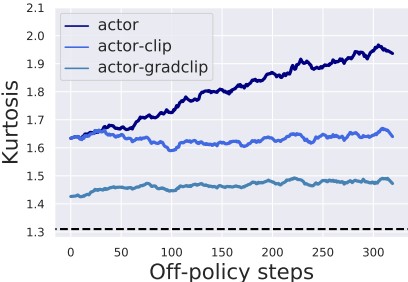 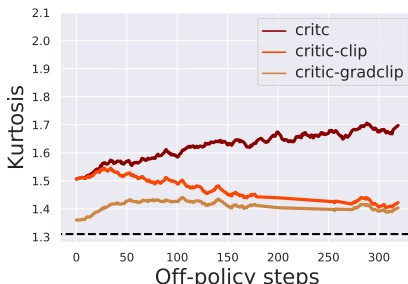

Figure 3: **Heavy-tailedness in PPO-NoClip with PPO-heuristics** applied progressively during off-policy steps, with kurtosis aggregated across 8 MuJoCo environments. For other estimators, see App. F. Dotted line represents the Kurtosis value for a Gaussian distribution. "-clip" denotes loss clipping on corresponding networks. "-gradclip" denotes both gradient clipping and loss clipping. Increases in Kurtosis implies an increase in heavy-tailedness. As training progresses during off-policy steps, the increased heavy-tailedness in actor and critic gradients is mitigated by PPO-heuristics.

During off-policy training, to understand the heavy-tailedness of actor gradients beyond the contributions from likelihood ratios, we inspect the actor gradients normalized by likelihood-ratios, i.e.,

$$\frac{\nabla_\theta \pi_\theta(a_t, s_t)/\pi_0(a_t, s_t)}{\pi_\theta(a_t, s_t)/\pi_0(a_t, s_t)} \hat{A}_{\pi_0}(s_t, a_t) = \nabla_\theta \log\left(\pi_\theta(a_t, s_t)\right) \hat{A}_{\pi_0}(s_t, a_t). \tag{5}$$

Note that the gradient expression in Eq. 5 is similar to on-policy actor gradients. Since we observe an increasing trend in heavy-tailedness of the actor gradients even during on-policy training, one might ask: does these gradients' heavy-tailedness increase during off-policy gradient updates? Recall that in PPO, we fix the value function at the beginning of off-policy training and pre-compute advantage estimates that will later be used throughout the training. Since the advantages were the primary factor dictating the increase during on-policy training, ideally, we should not observe any increase in the heavy-tailed behavior. Confirming this hypothesis, we show that the heavy-tailedness in this quantity indeed stays constant during the off-policy training (Fig. 2), i.e., $\nabla_\theta \log\left(\pi_\theta(a_t, s_t)\right) A_{\pi_0}(s_t, a_t)$ doesn't cause the increased heavy-tailed nature as long as $\pi_0$ is fixed.

Our findings from off-policy analysis strongly suggest that when the behavioral policy is held fixed, heavy-tailedness in the importance ratios $\rho$ is the fundamental cause. In addition, in Sec. 3.1, we showed that when importance-ratio's are 1 (i.e., the data on which the gradient step is taken is on-policy) advantages induce heavy-tailedness. With these two observations, we conclude that the scalars (either the likelihood-ratios or the advantage estimates) in the loss objective are the primary causes to the underlying heavy-tailedness in the gradients. Having analyzed key components dictating the heavy-tailed behavior, in App. J, we present two ablation experiments to test how heavy-tailedness in likelihood-ratios and advantage estimates individually contribute to the optimization issues in PPO leading to poor performance. Next, we illustrate how PPO clipping heuristics alleviate heavy-tailedness issues.

### 3.3 Explaining roles of various PPO objective optimizations

Motivated from our results from the previous sections, we now take a deeper look at how the core idea of likelihood-ratio clipping and auxiliary optimizations implemented in PPO and understand how they affect the heavy-tailedness during training. First, we make a key observation. Note that the PPO-clipping heuristics don't get triggered for the first gradient step taken (when a new batch of data is sampled). But rather these heuristics may alter the loss only when behavior policy is different from the policy that is being optimized. Hence, in order to understand the effects of clipping heuristics, we perform the following analysis on the off-policy gradients of the PPO-NoClip: At each update step on the agent trained with PPO-NoClip, we compute the gradients with progressively including optimizations from the standard PPO objective.

Our results demonstrate that both the likelihood-ratio clipping and value-function clipping in loss during training offset the enormous heavy-tailedness induced due to off-policy training (Fig. 3). Recall that by clipping the likelihood ratios and the value function, the PPO objective is discarding samples (i.e., replacing them with zero when) used for gradient aggregation. Since heavy-tailedness in the distribution of likelihood ratios is the central contributing factor during off-policy training, by truncating likelihood-ratios $\rho_t$ which lie outside $(1 - \epsilon, 1 + \epsilon)$ interval, PPO is primarily mitigating heavy-tailedness in actor gradients. Similarly, by rejecting samples from the value function loss which lie outside an $\epsilon$ boundary of a fixed *target* estimate, the heuristics alleviate the slight heavy-tailed nature induced with off-policy training in the critic network.

While these PPO heuristics alleviate the heavy-tailedness induced with off-policy training, the effects of PPO clipping optimizations on mitigating heavy-tailedness induced during on-policy updates are far less clear. Since none of these heuristics directly target the outliers present in the advantage-estimates (the primary cause of increasing heavy-tailedness throughout training), we believe that our findings can guide a development of fundamentally stable RL algorithms. In App. J.1, we present a preliminary result in this direction demonstrate how fixing heavy-tailedness in advantage estimates can improve an agent's performance.

## 4 MITIGATING HEAVY-TAILEDNESS WITH ROBUST GRADIENT ESTIMATION

Motivated by our analysis showing that the gradients in PPO-NOCLIP exhibit heavy-tailedness that increases during off-policy training, we propose an alternate method of gradient aggregation—using the gradient estimation framework from Prasad et al. (2018)—that is better suited to the heavy-tailed estimation paradigm than the sample mean. To support our hypothesis that addressing the primary benefit of PPO's various clipping heuristics lies in mitigating this heavy-tailedness, we aim to show that equipped with our robust estimator, PPO-NOCLIP can achieve comparable results to state-of-the-art PPO implementations, even with the clipping heuristics turned off.

We now consider robustifying PPO-NOCLIP (policy gradient with importance sampling but without any trust-region enforcing steps or PPO clipping tricks). Informally, for gradient distributions which do not enjoy Gaussian-like concentration, the empirical-expectation-based estimates of the gradient do not necessarily point in the right descent direction, leading to bad solutions. To this end, we leverage a robust mean aggregation technique called Geometric Median-Of-Means (GMOM) due to Minsker et al. (2015). In short, we first split the samples into non-overlapping subsamples and estimate the sample mean of each.

The GMOM estimator is then given by the geometric median-of-means of the subsamples. Formally, let $\{x_1, \ldots, x_n\} \in R$ be $n$ i.i.d. random variables sampled from a distribution $\mathcal{D}$. Then the GMOM estimator for estimating the mean can be described as follows: Partition the $n$ samples into b blocks $B_1, \ldots, B_b$, each of size $\lfloor n/b \rfloor$. Compute sample means in each block, i.e., $\{\hat{\mu}_1, \ldots, \mu_b\}$, where $\hat{\mu}_i = \sum_{x_j \in B_i} x_j / |B_i|$. Then the GMOM estimator $\hat{\mu}_{\text{GMOM}}$ is given by the *geometric median* of $\{\hat{\mu}_1, \ldots, \mu_b\}$ defined as follows: $\hat{\mu}_{\text{GMOM}} = \arg\min_\mu \sum_{i=1}^b \|\mu - \hat{\mu}_i\|_2$. Algorithm 2 in Appendix C presents the algorithm formally along with the Weiszfeld's algorithm used for computing the approximate geometric median.

---
**Algorithm 1** BLOCK-GMOM

**input** : Samples $S = \{x_1, \ldots, x_n\}$, number of blocks $b$, Model optimizer $\mathcal{O}_G$, $b$ block optimizers $\mathcal{O}_B$, network $f_\theta$, loss $\ell$
1: Partition S into b blocks $B_1, \ldots B_b$ of equal size.
2: **for** $i$ in $1 \ldots b$ **do**
3: $\quad \hat{\mu}_i = \mathcal{O}_B^{(i)} \left( \sum_{x_j \in B_i} \nabla_\theta \ell(f_\theta, x_j) / |B_i| \right)$
4: **end for**
5: $\hat{\mu}_{\text{GMOM}} = \mathcal{O}_G \left( \text{WEISZFELD}(\hat{\mu}_1, \ldots, \hat{\mu}_b) \right)$.
**output** : Gradient estimate $\hat{\mu}_{\text{GMOM}}$

---

GMOM has been shown to have several favorable properties when used for statistical estimation in heavy-tailed settings. Intuitively, GMOM reduces the effect of outliers on a mean estimate by taking a intermediate mean of blocks of samples and then computing the geometric median of those block means. The robustness comes from the additional geometric median step where a small number of samples with large norms would not affect a GMOM estimate as much as they would a sample mean. More formally, given $n$ samples from a heavy-tailed distribution, the GMOM estimate concentrates better around the true mean than the sample mean which satisfies the following:

**Theorem 1** (Minsker et al. (2015)). *Suppose we are given $n$ samples $\{x_i\}_{i=1}^n$ from a distribution with mean $\mu$ and covariance $\Sigma$. Assume $\delta > 0$. Choose the number of blocks $b = 1 + \lfloor 3.5 \log(1/\delta) \rfloor$. Then, with probability at least $1 - \delta$,*

$$\|\mu_{\text{GMOM}} - \mu\|_2 \lesssim \sqrt{\frac{trace(\Sigma) \log(1/\delta)}{n}} \;\; and \;\; \left\| \frac{1}{n} \sum_{i=1}^n x_i - \mu \right\|_2 \gtrsim \sqrt{\frac{trace(\Sigma)}{n\delta}} \;.$$

When applying stochastic gradient descent or its variants in deep learning, one typically backpropagates the mean loss, avoiding computing per-sample gradients. However, computing GMOM requires per-sample gradients. To this end, we propose a simple (but novel) variant of GMOM called BLOCK-GMOM which avoids the extra sample-size dependent computational penalty of calculating sample-wise gradients. Notice that in Theorem 1, the number of blocks required to obtain the guarantee with high probability is independent of the sample size, i.e., we just need a constant (dependent on $\delta$) number of blocks to compute GMOM. To achieve this, instead of calculating sample-wise gradients, we compute block-wise gradients by backpropagating on sample-mean aggregated loss for each block. Moreover, such an implementation not only increases efficiency but also allows incorporating adaptive optimizers for individual blocks. Algorithm 1 presents the overall BLOCK-GMOM.

### 4.1 RESULTS ON MUJOCO ENVIRONMENT

We perform experiments on 8 MuJoCo (Todorov et al., 2012) continuous control tasks. To use BLOCK-GMOM aggregation with PPO-NOCLIP, we extract actor-network and critic-network gradients at each step and separately run the Algorithm 1 on both the networks. For our experiments, we use SGD as $\mathcal{O}_B$ and Adam as $\mathcal{O}_G$ and refer to this variant of PPO-NOCLIP as ROBUST-PPO-NOCLIP. We compare the performances of PPO, PPO-NOCLIP, and RO-BUST-PPO-NOCLIP, using hyperparameters that are tuned individually for each method but held fixed across all tasks (Table 1). For 7 tasks, we observe significant improvements with RO-BUST-PPO-NOCLIP over PPO-NOCLIP and performance close to that achieved by PPO (with all clipping heuristics enabled) (Fig. 4). Although we do not observe improvements over PPO, we believe that this result corroborates our conjecture that PPO heuristics primarily aim to offset the heavy-tailedness induced with training.

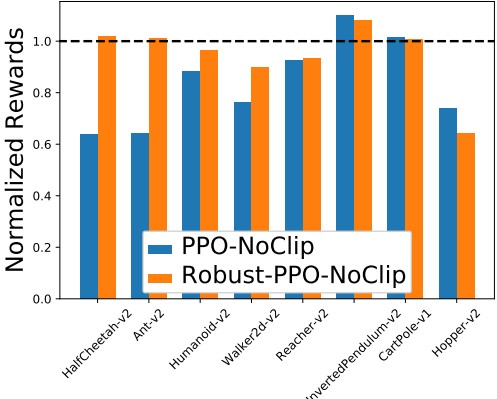

Figure 4: **Normalized rewards for ROBUST-PPO-NOCLIP and PPO-NOCLIP**. Normalized w.r.t. the max reward obtained with PPO (with all heuristics enabled) and performance of a random agent. (See App G for reward curves on individual environment.)

## 5 RELATED WORK

Studying the behavior of SGD, Simsekli et al. (2019) questioned the Gaussianity of SGD noise, highlighting its *heavy-tailed* nature. Subsequently, there has been a growing interest in understanding the nature of SGD noise. More recently, other works (Şimşekli et al., 2020; Zhang et al., 2019b) explore the heavy-tailed behavior of SGD noise, with emphasis on tasks that require adaptive methods. In particular, Zhang et al. (2019b) studied the nature of gradients in natural language processing (e.g., BERT-pretraining). Later work (Panigrahi et al., 2019) studied results through Gaussianity tests on random projections. They showed that at least early on in training, gradients remain near Gaussian for larger batch sizes. Some recent work has also made progress towards understanding the effectiveness of gradient clipping in convergence (Zhang et al., 2019b;a; Şimşekli et al., 2020). However, its consequences modulo the bias- vs variance trade-off are not completely understood.

On the RL side, a large-scale study of PPO was presented in (Ilyas et al., 2018), examining the extent that the PPO gradients align to the true underlying gradient. Likewise, Engstrom et al. (2019) provide a thorough study of the code-level heuristics used in trust region methods and indicate the

necessity of such heuristics in obtaining strong model performance. Chung et al. (2020) highlighted the impacts of stochasticity on the optimization process in policy gradients. In simple MDPs, authors showed that larger higher moments with fixed variance lead to improved exploration. This aligns with one view of heavy-tailedness in supervised learning where Simsekli et al. (2019) conjectured that heavy-tailedness in gradients can improve generalization.

However, we hypothesize that in deep RL where the optimization process is known to be brittle (Henderson et al., 2018; 2017; Engstrom et al., 2019; Ilyas et al., 2018), heavy-tailedness can cause heightened instability than help in efficient exploration. This perspective aligns with another line of work (Zhang et al., 2019b;a) where authors demonstrate that heavy-tailedness can cause instability in the learning process in deep models. Indeed with ablation experiments in Appendix J, we show that increasing heavy-tailedness in likelihood ratios hurt the agent's performance, and mitigating heavy-tailedness in advantage estimates improves learning dynamics and hence the agent's performance. Chung et al. (2020) further pointed out the importance of a careful analysis of stochasticity in updates to understand the optimization process of policy gradient algorithms. We consider that our work is a stepping stone towards analyzing stochasticity beyond variance.

Bubeck et al. (2013) studied the stochastic multi-armed bandit problem when the reward distribution is heavy-tailed. The authors designed a robust version of the classical Upper Confidence Bound algorithm by replacing the empirical average of observed rewards with robust estimates obtained via the univariate median-of-means estimator (Nemirovski & Yudin, 1983) on the observed sequence of rewards. Medina & Yang (2016) extended this approach to the problem of linear bandits under heavy-tailed noise.

## 6 CONCLUSION

In this paper, we empirically characterized PPO's gradients, demonstrating that they become more heavy-tailed as training proceeds. Our detailed analysis showed that at on-policy steps, the heavy-tailed nature of the gradients is primarily attributable to the multiplicative advantage estimates. On the other hand, we observed that during off-policy training, the heavy-tailedness of the likelihood ratios of the surrogate reward function exacerbates the observed heavy-tailedness.

Subsequently, we examined PPO's clipping heuristics, showing that they serve primarily to offset the heavy-tailedness induced by off-policy training. Thus motivated, we showed that a robust estimation technique could effectively replace all three of PPO's clipping heuristics: likelihood-ratio clipping, value loss clipping, and gradient clipping.

In future work, we plan to conduct similar analysis on gradients for other RL algorithms such as deep Q-learning. We believe that progress in this direction can significantly impact algorithm development for reinforcement learning.

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

# A DETAILED BACKGROUD

We define a Markov Decision Process (MDP) as a tuple $(\mathcal{S}, \mathcal{A}, R, \gamma, P)$, where $\mathcal{S}$ represent the set of environments states, $\mathcal{A}$ represent the set of agent actions, $R : \mathcal{S} \times \mathcal{A} \to \mathbb{R}$ is the reward function, $\gamma$ is the discount factor, and $P : \mathcal{S} \times \mathcal{A} \times \mathcal{S} \to \mathbb{R}$ is the state transition probability distribution. The goal in reinforcement learning is to learn a policy $\pi_\theta : \mathcal{S} \times \mathcal{A} \to [0, 1]$, parameterized by $\theta$, such that the expected cumulative discounted reward (known as returns) is maximized. Formally,

$$\pi^* = \operatorname*{argmax}_\pi \mathbb{E}_{a_t \sim \pi(\cdot|s_t), s_{t+1} \sim P(\cdot|s_t, a_t)} \left[ \sum_{t=0}^\infty \gamma^t R(s_t, a_t) \right] . \tag{6}$$

Policy gradient methods directly optimize a paraterized policy function (also known as *actor network*). The central idea behind policy gradient methods is to perform stochastic gradient ascent on expected return (Eq. 6) to learn parameters $\theta$. Under mild conditions (Sutton et al., 2000), the gradient of the Eq. 6 can be written as

$$\nabla_\theta J(\theta) = \mathbb{E}_{\tau \sim \pi_\theta} \left[ \sum_{t=0}^\infty \gamma^t R(s_t, a_t) \nabla_\theta \log(\pi_\theta(a_t|s_t)) \right] ,$$

where $\tau \sim \pi_\theta$ are trajectories sampled according to $\pi_\theta(\tau)$ and $J(\theta)$ is the objective maximised in Eq. 6. With the observation that action $a_t$ only affects the reward from time $t$ onwards, we re-write the objective $J(\theta)$, replacing returns using the Q-function, i.e., the expected discounted reward after taking an action $a$ at state $s$ and following $\pi_\theta$ afterwards. Mathematically, $Q_{\pi_\theta}(s, a) = \mathbb{E}_{\tau \sim \pi_\theta} \left[ \sum_{k=0}^\infty \gamma^k R(s_{t+k}, a_{t+k}) | a_t = a, s_t = s \right]$. Using the Q-function, we can write the gradient of the objective function as

$$\nabla_\theta J(\theta) = \mathbb{E}_{\tau \sim \pi_\theta} \left[ \sum_{t=0}^\infty Q_{\pi_\theta}(s_t, a_t) \nabla_\theta \log(\pi_\theta(a_t|s_t)) \right] .$$

However, the variance in the above expectation can be large, which raises difficulties for estimating the expectation empirically. To reduce the variance of this estimate, a baseline is subtracted from the Q-function—often the value function or expected cumulative discounted reward starting at a certain state and following a given policy i.e., $V_{\pi_\theta}(s) = \mathbb{E}_{\tau \sim \pi_\theta} \left[ \sum_{k=0}^\infty \gamma^k R(s_{t+k}, a_{t+k}) | s_t = s \right]$. The network that estimates the value function is often referred to as *critic*. Define $A_{\pi_\theta}(s_t, a_t) = Q_{\pi_\theta}(s_t, a_t) - V_{\pi_\theta}(s_t)$ as the *advantage* of performing action $a_t$ at state $s_t$. Incorporating an advantage function, the gradient of the objective function can be written:

$$\nabla_\theta J(\theta) = \mathbb{E}_{\tau \sim \pi_\theta} \left[ \sum_{t=0}^\infty A_{\pi_\theta}(s_t, a_t) \nabla_\theta \log(\pi_\theta(a_t|s_t)) \right] . \tag{7}$$

Eq. 7 is the näive actor-critic objective and is used by A2C.

**Trust region methods and PPO.** Since directly optimizing the cumulative rewards can be challenging, modern policy gradient optimization algorithms often optimize a surrogate reward function in place of the true reward. Most commonly, the surrogate reward objective includes a likelihood ratio to allow importance sampling from a behavior policy $\pi_0$ while optimizing policy $\pi_\theta$, such as the surrogate reward used by Schulman et al. (2015a):

$$\max_\theta \mathbb{E}_{(s_t, a_t) \sim \pi_0} \left[ \frac{\pi_\theta(a_t, s_t)}{\pi_0(a_t, s_t)} \hat{A}_{\pi_0}(s_t, a_t) \right] , \tag{8}$$

where $\hat{A}_\pi = \frac{A_\pi - \mu(A_\pi)}{\sigma(A_\pi)}$ (we refer to this as the *normalized advantages*). However, the surrogate is indicative of the true reward function only when $\pi_\theta$ and $\pi_0$ are close in distribution. Different policy gradient methods (Schulman et al., 2015a; 2017; Kakade, 2002) attempt to enforce the closeness in different ways. In Natural Policy Gradients (Kakade, 2002) and Trust Region Policy Optimization (TRPO) (Schulman et al., 2015a), authors utilize a conservation policy iteration with an explicit divergence constraint which provides provable lower bounds guarantee on the improvements of the parameterized policy. On the other hand, PPO (Schulman et al., 2017) implements a clipping

heuristic on the likelihood ratio of the surrogate reward function to avoid excessively large policy updates. Specifically, PPO optimizes the following objective:

$$\max_\theta \mathbb{E}_{(s_t, a_t) \sim \pi_0} \left[ \min \left( \text{clip}(\rho_t, 1 - \epsilon, 1 + \epsilon) \hat{A}_{\pi_0}(s_t, a_t), \rho_t \hat{A}_{\pi_0}(s_t, a_t) \right) \right], \tag{9}$$

where $\rho_t \colon = \frac{\pi_\theta(a_t, s_t)}{\pi_0(a_t, s_t)}$. We refer to $\rho_t$ as *likelihood-ratios*. Due to a minimum with the unclipped surrogate reward, the PPO objective acts as a pessimistic bound on the true surrogate reward. As in standard PPO implementation, we use Generalized Advantage Estimation (GAE) (Schulman et al., 2015b). Moreover, instead of fitting the value network via regression to target values:

$$L^V = (V_{\theta_t} - V_{targ})^2, \tag{10}$$

standard implementations fit the value network with a PPO-like objective:

$$L^V = \max \left[ \left(V_{\theta_t} - V_{targ}\right)^2, \left( \text{clip}\left(V_{\theta_t}, V_{\theta_{t-1}} - \varepsilon, V_{\theta_{t-1}} + \varepsilon\right) - V_{targ} \right)^2 \right], \tag{11}$$

where $\epsilon$ is the same value used to clip probability raitos in PPO's loss function (Eq. **??**). PPO uses the following training procedure: At any iteration $t$, the agent creates a clone of the current policy $\pi_{\theta_t}$ which interacts with the environment to collects rollouts $\mathcal{S}$ (i.e., state-action pair $\{(s_i, a_i)\}_{i=1}^N$). Then the algorithm optimizes the policy $\pi_\theta$ and value function for a fixed $K$ gradient steps on the sampled data $\mathcal{S}$. Since at every iteration the first gradient step is taken on the same policy from which the data was sampled, we refer to these gradient updates as *on-policy* steps. And as for the remaining $K - 1$ steps, the sampling policy differs from the current agent, we refer to these updates as *off-policy* steps.

Throughout the paper, we consider a stripped-down variant of PPO (denoted PPO-NOCLIP) that consists of policy gradient with importance weighting (Eq. 8), but has been simplified as follows: i) no likelihood-ratio clipping, i.e., no *objective function clipping*; ii) value network optimized via regression to target values (Eq. 10) without *value function clipping*; and iii) no *gradient clipping*. Overall PPO-NOCLIP uses the following objective:

$$\max_\theta \mathbb{E}_{(s_t, a_t) \sim \pi_0} \left[ \frac{\pi_\theta(a_t, s_t)}{\pi_0(a_t, s_t)} \hat{A}_{\pi_0}(s_t, a_t) - c(V_{\theta_t} - V_{targ})^2 \right].$$

where $c$ is a coefficient of the value function loss (tune as a hyperparameter). Moreover, no gradient clipping is incorporated in PPO-NOCLIP. One may argue that since PPO-NOCLIP removes the clipping heuristic from PPO, the unconstrained maximization of Eq. 1 may lead to excessively large policy updates. In App. I, we empirically justify the use of Eq. 1 by showing that with the small learning rate used in our experiments (optimal hyperparameters in Table 1), PPO-NOCLIP maintains a KL based trust-region like PPO throughout the training. We elaborate this in App. I.

## B   DETAILS ON ESTIMATORS

We now formalize our setup for studying the distribution of gradients. Throughout the paper, we use the following definition of the heavy-tailed property:

**Definition 2** (Resnick (2007)). *A non-negative random variable $w$ is called* heavy-tailed *if its tail probability $F_w(t) \colon = P(w \geq t)$ is asymptotically equivalent to $t^{-\alpha^*}$ as $t \to \infty$ for some positive number $\alpha^*$. Here $\alpha^*$ determines the heavy-tailedness and $\alpha^*$ is called tail index of $w$.*

For a heavy-tailed distribution with index $\alpha^*$, its $\alpha$-th moment exist only if $\alpha < \alpha^*$, i.e., $\mathbb{E}[w^\alpha] < \infty$ iff $\alpha < \alpha^*$. A value of $\alpha^* = 1.0$ corresponds to a Cauchy distribution and $\alpha^* = \infty$ (i.e., all moments exist) corresponds to a Gaussian distribution. Intuitively, as $\alpha^*$ decreases, the central peak of the distribution gets higher, the valley before the central peak gets deeper, and the tails get heavier. In other words, the lower the tail-index, the more heavy-tailed the distribution. However, in the finite sample setting, estimating the tail index is notoriously challenging (Simsekli et al., 2019; Danielsson et al., 2016; Hill, 1975).

In this study, we explore three estimators as heuristic measures to understand heavy tails and non-Gaussianity of gradients.

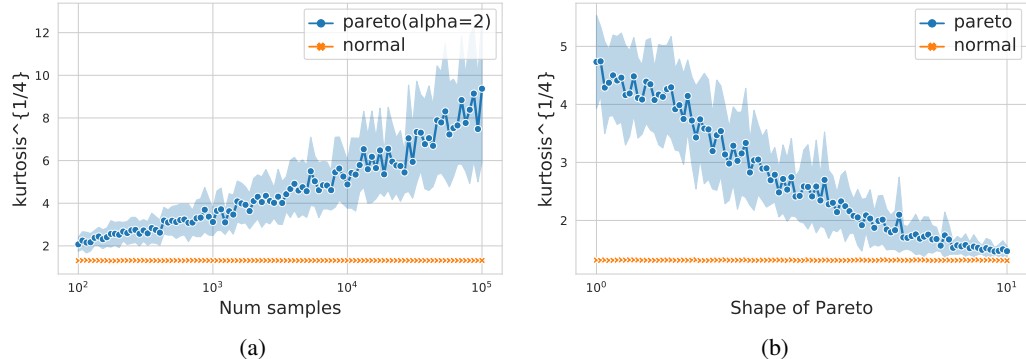

(a)                                              (b)

Figure 5: Kurtosis plots. Analysis on norms of 100-dimensional vectors such that each coordinate is sampled iid from Pareto distribution or normal distribution. (a) Variation in kurtosis ($\kappa^{1/4}$) as the sample size is varied for samples from normal distribution and Pareto with tail index 2 (i.e, $\alpha = 2$). (b) Variation in kurtosis ($\kappa^{1/4}$) as the shape of Pareto is varied at fix sample size.

- *Alpha-index estimator.* This estimator was proposed in Mohammadi et al. (2015) for symmeteric $\alpha$-stable distributions and was used by Simsekli et al. (2019) to understand the noise behavior of SGD. This estimator is derived under the (strong) assumption that the stochastic Gradient Noise (GN) vectors are coordinate-wise independent and follow a symmetric alpha-stable distribution. Formally, let $\{X_i\}_{i=1}^N$ be a collection of $N = mn$ (centered) random variables. Define $Y_i = \sum_{j=1}^m X_{j+(i-1)m}$ for $i \in [n]$. Then, the estimator is given by

$$\frac{1}{\alpha} : = \frac{1}{\log m} \left( \frac{1}{n} \sum_{i=1}^n \log |Y_i| - \frac{1}{n} \sum_{i=1}^n N \log |X_i| \right) .$$

  Instead of treating each co-ordinate of gradient noise as an independent scalar, we use these estimators on gradient norms. With alpha-index estimator, smaller alpha-index value signify higher degree of heavy-tailedness.

- *Anderson-Darling test* (Anderson & Darling, 1954) on random projections of GN to perform Gaussianity testing. Panigrahi et al. (2019) proposed the Gaussianity test on the projections of GN along 1000 random directions. Their estimate is then the fraction of directions accepted by the Anderson Darling test. While this estimator is informative about the Gaussian behavior, it is not useful to quantify and understand the trends of heavy-tailedness if the predictor nature is non-Gaussian.

- To our knowledge, the deep learning literature has only exploredthese two estimators for analyzing the heavy-tailed nature of gradients. *(iii)* Finally, in our work, we propose using kurtosis *Kurtosis.* To quantify the heavy-tailedness relative to a normal distribution, we measure kurtosis (fourth standardized moment) of the gradient norms. Given samples $\{X_i\}_{i=1}^N$, the kurtosis $\kappa$ is given by

$$\kappa = \frac{\sum_{i=1}^N (X_i - \bar{X})^4 / N}{\left( \sum_{i=1}^N (X_i - \bar{X})^2 / N \right)^2} ,$$

  where $\bar{X}$ is the empirical mean of the samples.

## B.1 SYNTHETIC STUDY

In Figure 5, we show the trends with varying tail index and sample sizes. Clearly as the tail-index increases, i.e., the shape parameter increases, the kurtosis decreases (signifying its correlation to capture tail-index). Although for tail-index smaller than 4 the kurtosis is not defined, we plot empirical kurtosis and show its increasing trend sample size. We fix the tail index of Pareto at 2 and plot

finite sample kurtosis and observe that it increases almost exponentially with the sample size. These two observations together hint that kurtosis is a neat surrogate measure for heavy-tailedness.

## C  GMOM ALGORITHM

---

**Algorithm 2** GMOM

**input** : Samples $S = \{x_1, \ldots, x_n\}$, number of blocks $b$
1: $m = \lfloor n/b \rfloor$.
2: **for** $i$ in $1 \ldots b$ **do**
3:     $\hat{\mu}_i = \sum_{j=0}^{m} x_{j+i*m}/$ m.
4: **end for**
5: $\hat{\mu}_{\text{GMOM}} = \text{WEISZFELD}(\hat{\mu}_1, \ldots, \hat{\mu}_b)$.
**output** : Estimate $\hat{\mu}_{\text{GMOM}}$

---

**Algorithm 3** WEISZFELD

**input** : Samples $S = \{\mu_1, \ldots, \mu_b\}$, number of blocks $b$
1: Initialize $\mu$ arbitrarily.
2: **for** iteration $\leftarrow 1, \ldots, n$ **do**
3:     $d_j := \frac{1}{\|\mu - \mu_j\|_2}$ for $j$ in $1, \ldots, b$.
4:     $\mu := \left( \sum_{j=1}^{b} \mu_j d_j \right) / \left( \sum_{j=1}^{b} d_j \right)$
5: **end for**
**output** : Estimate $\mu$

---

## D  EXPERIMENTAL SETUP FOR GRADIENT DISTRIBUTION STUDY

Recall that PPO uses the following training procedure: At any iteration $t$, the agent creates a clone of the current policy $\pi_{\theta_t}$ which interacts with the environment to collects rollouts $\mathcal{S}$ (i.e., state-action pair $\{(s_i, a_i)\}_{i=1}^{N}$). Then the algorithm optimizes the policy $\pi_\theta$ and value function for a fixed $K$ gradient steps on the sampled data $\mathcal{S}$. Since at every iteration the first gradient step is taken on the same policy from which the data was sampled, we refer to these gradient updates as **on-policy steps**. And as for the remaining $K - 1$ steps, the sampling policy differs from the current agent, we refer to these updates as **off-policy steps**. For all experiments, we aggregate our estimators across 30 seeds and 8 environments. We do this by first computing the estimators for individual experiments and then taking the sample mean across all runs. We now describe the exact experimental details.

In all of our experiments, for each gradient update, we have a batch size of 64. Hence for an individual estimate, we aggregate over 64 samples (batch size in experiments) to compute our estimators. For Anderson Darling test, we use 100 random directions to understand the behavior of stochastic gradient noise.

**On-policy heavy-tailed estimation.** At every on-policy gradient step (i.e. first step on newly sampled data), we freeze the policy and value network, and save the sample-wise gradients of the actor and critic objective. The estimators are calculated at every tenth on-policy update throughout the training.

**Off-policy heavy-tailed estimation** At every off-policy gradient step (i.e. the gradient updates made on a fixed batch of data when the sampling policy differs from the policy being optimized), we freeze the policy and value network, and save the sample-wise gradients of the actor and critic objective. Then at various stages of training, i.e., initialization, $50\%$ max reward and max reward (which corresponds to different batches of sampled data), we fix the collected trajectories and collect sample-wise gradients for the 320 steps taken. We now elaborate the exact setup with one instance, at $50\%$ of the maximum reward. First, we find the training iteration where the agent achieves approximately $50\%$ of the maximum reward individually for each environment. Then at this training iteration, we freeze the policy and value network and save the sample-wise gradients of the actor and critic objective for off-policy steps.

**Analysis of PPO-NOCLIP with progressively applying PPO heuristics**. We compute the gradients for the off-policy steps taken with the PPO-NOCLIP objective as explained above. Then at each gradient step, we progressively add heuristics from PPO and re-compute the gradients for analysis. Note that we still always update the value and policy network with PPO-NOCLIP objective gradients.

# E TRENDS WITH ADVANTAGES

## E.1 KURTOSIS FOR RETURNS, VALUE ESTIMATE AND ADVANTAGES GROUPED WITH SIGN

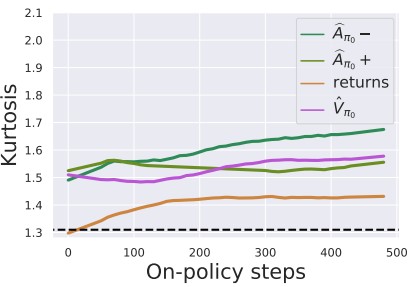

Figure 6: Heavy-tailedness in advantages grouped by their sign, rewards and value estimates. Clearly, as the training progresses the negative advantages become heavy-tailed. For returns, we observe an initial slight increase in the heavy-tailedness which quickly plateaus to a small magnitude of heavytailedness. The heavytailedness in the value estimates and positive advantages remain almost constant throughout the training.

## E.2 HEAVY-TAILEDNESS IN A2C AND PPO IN ONPOLICY ITERATIONS

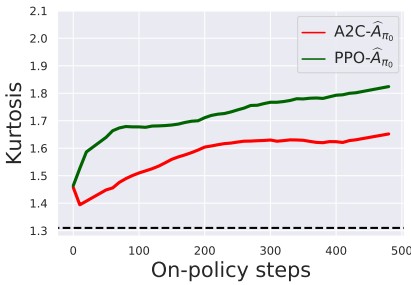

Figure 7: Heavy-tailedness in advantages for A2C and PPO during on-policy iterates. Clearly, as the training progresses heavy-tailedness in PPO advantages increases rapidly when compared with A2C advantages. The observed behavior arises to the off-policy training of the agent in PPO. This explains why we observe heightened heavy-tailedness in PPO during onpolicy iterations in Fig 1(a).

## E.3 HISTOGRAMS OF ADVANTAGES ON HALFCHEETAH OVER TRAINING ITERATIONS

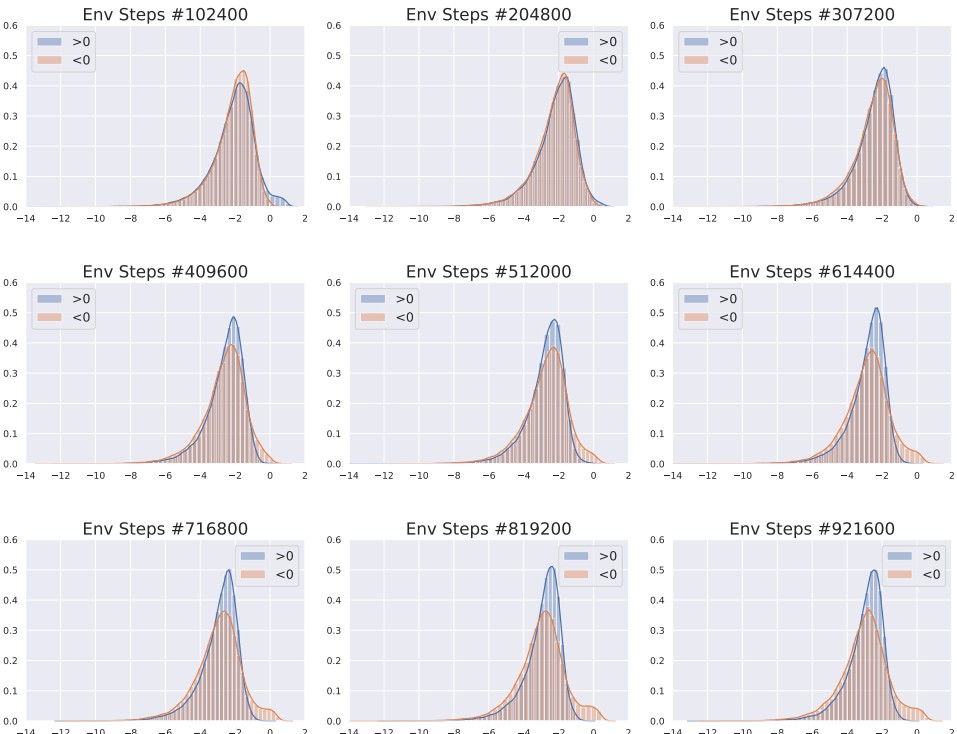

Figure 8: Distribution of $\log(|A_{\pi_\theta}|)$ over training grouped by sign of $\log(|A_{\pi_\theta}|)$ for HalfCheetah-v2 . To elaborate, we collect the advantages and separately plot the grouped advantages with their sign, i.e., we draw histograms separately for negative and positive advantages. As training proceeds, we clearly observe the increasing heavy-tailed behavior in negative advatanges as captured by the higher fraction of $\log(|A_{\pi_\theta}|)$ with large magnitude. Moreover, the histograms for positive advantages (which resembel Gaussain pdf) stay almost the same throughout training. This highlights the particular heavy-tailed (outlier-rich) nature of negative advantages corroborating our experiments with kurtosis and tail-index estimators.

# F ANALYSIS WITH OTHER ESTIMATORS

## F.1 ON-POLICY GRADIENT ANALYSIS

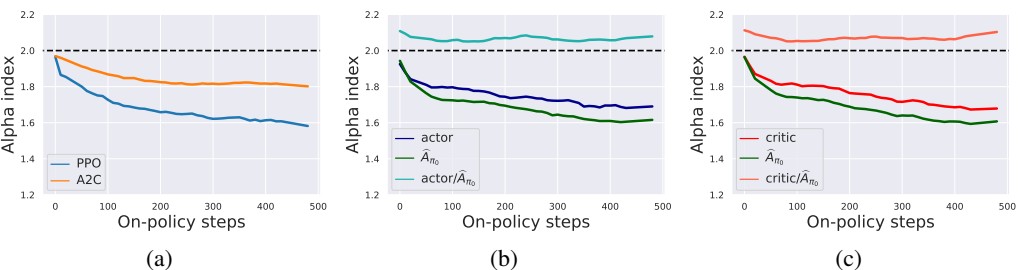

Figure 9: **Heavy-tailedness in PPO during on-policy iterations**. All plots show mean alpha index aggregated over 8 MuJoCo environments. A decrease in alpha-index implies an increase in heavy-tailedness. (a) Alpha index vs on-policy iterations for A2C and PPO. Evidently, as training proceeds, the gradients become more heavy-tailed for both the methods. (b) Alpha index vs on-policy iterations for actor networks in PPO. (c) Alpha index vs on-policy iterations for critic networks in PPO. Both critic and actor gradients become more heavy-tailed on-policy steps as the agent is trained. Note that as the gradients become more heavy-tailed, we observe a corresponding increase of heavy-tailedness in the advantage estimates ($\hat{A}_{\pi_0}$) . However, "actor/$\hat{A}_{\pi_0}$" and "critic/$\hat{A}_{\pi_0}$" (i.e., actor or critic gradient norm divided by GAE estimates) remain light-tailed throughout the training.

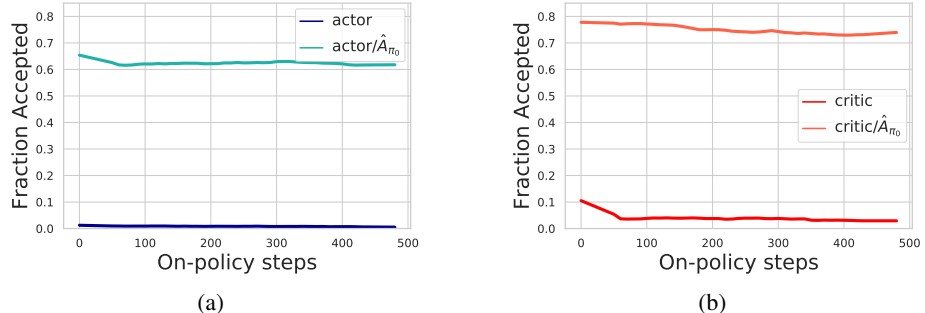

Figure 10: **Heavy-tailedness in PPO during on-policy iterations**. All plots show mean fraction of directions accepted by Anderon-Darling test over 8 MuJoCo environments. A higher accepted fraction indicates a Gaussian behavior. (b) Fraction accepted vs on-policy iterations for actor networks in PPO. (c) Fraction accepted vs on-policy iterations for critic networks in PPO. Both critic and actor gradients remain non-Gaussian as the agent is trained. However, "actor/$\hat{A}_{\pi_0}$" and "critic/$\hat{A}_{\pi_0}$" (i.e., actor or critic gradient norm divided by GAE estimates) have fairly high fraction of directions accepted, hinting their Gaussian nature.

### F.2    OFF-POLICY GRADIENT ANALYSIS

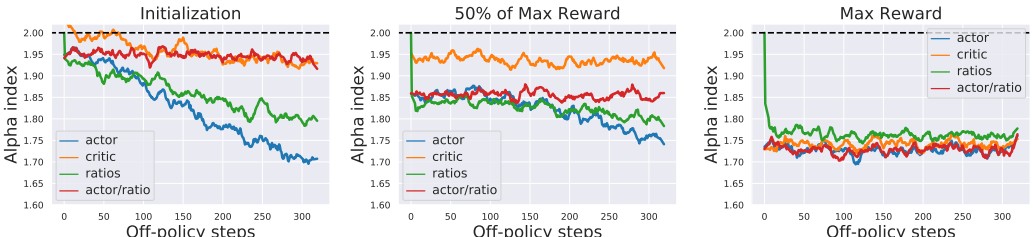

Figure 11: **Heavy-tailedness in PPO-NOCLIP during off-policy steps** at various stages of training iterations in MuJoCo environments. All plots show mean alpha index aggregated over 8 Mujoco environments. A decrease in alpha index implies an increase in heavy-tailedness. As off-policyness increases, the actor gradients get substantially heavy-tailed. This trend is corroborated by the increase of heavy-tailedness in ratios. Moreover, consistently we observe that the heavy-tailedness in "actor/ratios" stays constant. While initially during training, the heavy-tailedness in the ratio's increases substantially, during later stages the increase tapers off. The overall increase across training iterations is explained by the induced heavy-tailedness in the advantage estimates (cf. Sec. 3.1).

# G HYPERPARAMETER SETTINGS AND REWARDS CURVES ON INDIVIDUAL ENVIORNMENTS

| Hyperparameter | Values |
| --- | --- |
| Steps per PPO iteration | 2048 |
| Number of minibatches | 32 |
| PPO learning rate | 0.0003 |
| ROBUST-PPO-NOCLIP learning rate | 0.00008 |
| PPO-NOCLIP learning rate | 0.00008 |
| Discount factor $\gamma$ | 0.99 |
| GAE parameter $\lambda$ | 0.95 |
| Entropy loss coefficient | 0.0 |
| PPO value loss coefficient | 2.0 |
| ROBUST-PPO-NOCLIP value loss coefficient | 2.0 |
| PPO-NOCLIP value loss coefficient | 2.0 |
| Max global L2 gradient norm (only for PPO) | 0.5 |
| Clipping coefficient (only for PPO) | 0.2 |
| Policy epochs | 10 |
| Value epochs | 10 |
| GMOM number of blocks | 8 |
| GMOM Weiszfeld iterations | 100 |

Table 1: Hyperparameter settings. Sweeps were run over learning rates { 0.000025, 0.00005, 0.000075, 0.00008, 0.00009 , 0.0001, 0.0003, 0.0004 } and value loss coefficient { 0.1, 0.5, 1.0, 2.0, 10.0} with 30 random seeds per learning rate.

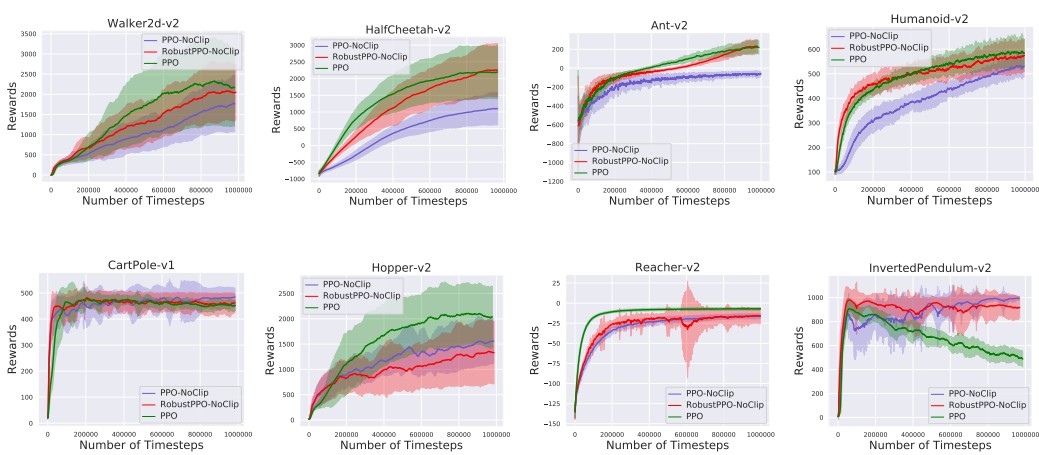

Figure 12: **Reward curves as training progresses in 8 different Mujoco Environments** aggregated across 30 random seeds and for hyperparameter setting tabulated in Table 1. The shaded region denotes the one standard deviation across seeds. We observe that except in Hopper-v2 environment, the mean reward with ROBUST-PPO-NOCLIP is significantly better than PPO-NOCLIP and close to that achieved by PPO with optimal hyperparameters. Aggregated results shown in Fig. 4.

# H ANALYSIS ON INDIVIDUAL ENVIORNMENTS.

Overall, in the figures below, we show that the trends observed in aggregated plots in Section 3 with Kurtosis hold true on individual environments. While the degree of heavy-tailedness varies in different environments, the trend of increase in heavy-tailedness remains the same.

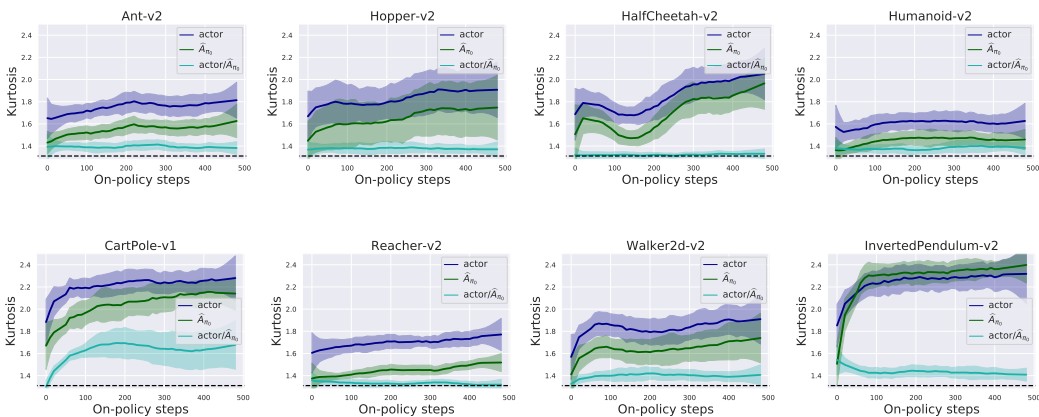

Figure 13: **Heavy-tailedness in actor gradients for PPO during on-policy steps** for 8 MuJoCo environments. All plots show mean and std of kurtosis aggregated over 30 random seeds. As the agent is trained, actor gradients become more heavy-tailed. Note that as the gradients become more heavy-tailed, we observe a corresponding increase of heavy-tailedness in the advantage estimates ($\hat{A}_{\pi_0}$). However, "actor/$\hat{A}_{\pi_0}$" (i.e., actor gradient norm divided by advantage) remain light-tailed throughout the training.

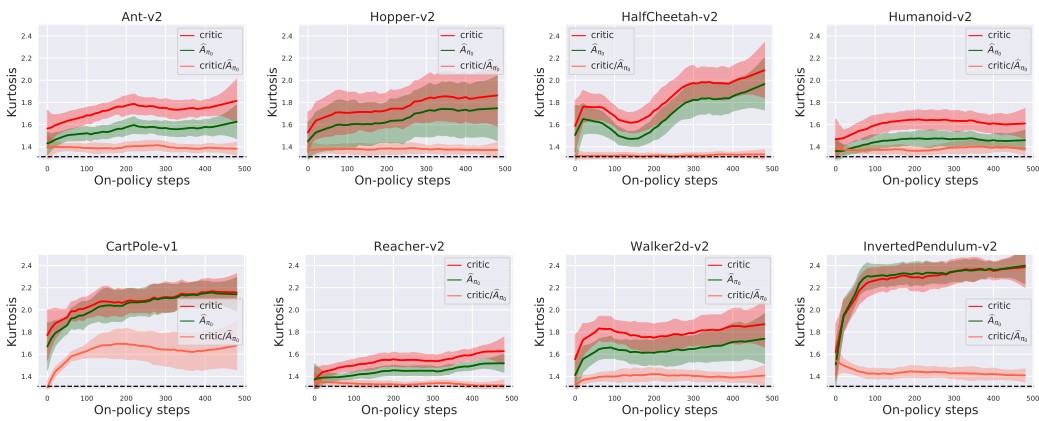

Figure 14: **Heavy-tailedness in critic gradients for PPO during on-policy steps** for 8 MuJoCo environments. All plots show mean and std of kurtosis aggregated over 30 random seeds. As the agent is trained, critic gradients become more heavy-tailed. Note that as the gradients become more heavy-tailed, we observe a corresponding increase of heavy-tailedness in the advantage estimates ($\hat{A}_{\pi_0}$). However, "critic/$\hat{A}_{\pi_0}$" (i.e., critic gradient norm divided by advantage) remain light-tailed throughout the training.

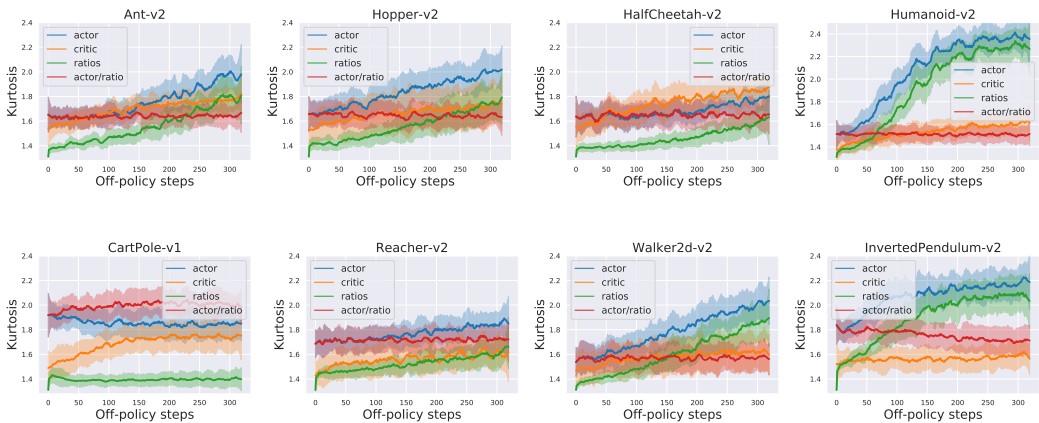

Figure 15: **Heavy-tailedness in PPO-NoClip during off-policy steps at Initialization** for 8 Mu-JoCo environments. All plots show mean and std of kurtosis aggregated over 30 random seeds. As off-policyness increases, the actor gradients get substantially heavy-tailed. This trend is corroborated by the increase of heavy-tailedness in ratios. Moreover, consistently we observe that the heavy-tailedness in "actor/ratios" stays constant. The trend in heavy-tailedness at later training iteration follow similar trends but the increase in heavy-tailedness tapers off. The overall increase across training iterations is explained by the induced heavy-tailedness in the advantage estimates (cf. Sec. 3.1).

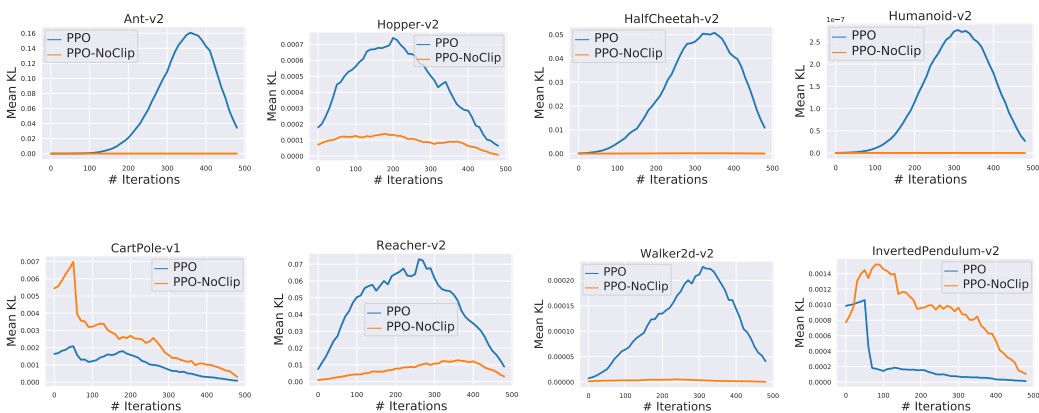

Figure 16: **KL divergence between current and previous policies with the optimal hyperparameters (parameters in Table 1)**. We measure the mean empirical KL divergence between the policy obtained at the end of off-policy training (after every 320 gradient steps) and the sampling policy at the beginning of every training iteration. The quantities are measured over the state-action pairs collected in the training step (Engstrom et al. (2019) observed similar results with both unseen data and training data). We observe that both the algorithms maintain a KL based trust region. The trend with KL divergence in PPO matches with the observations made in Engstrom et al. (2019) where they also observed that it peeks in halfway in training.

## I   MEAN KL DIVERGENCE BETWEEN CURRENT AND PREVIOUS POLICY

Enforcing a trust region is a core algorithmic property of PPO and TRPO. While the trust-region enforcement is not directly clear from the reward curves or heavy-tailed analysis, inspired by Engstrom et al. (2019), we perform an additional experiment to understand how this algorithmic property varies with PPO and our variant PPO-NoClip with optimal hyperparameters. In Fig 16, we measure mean KL divergence between successive policies of the agent while training with PPO

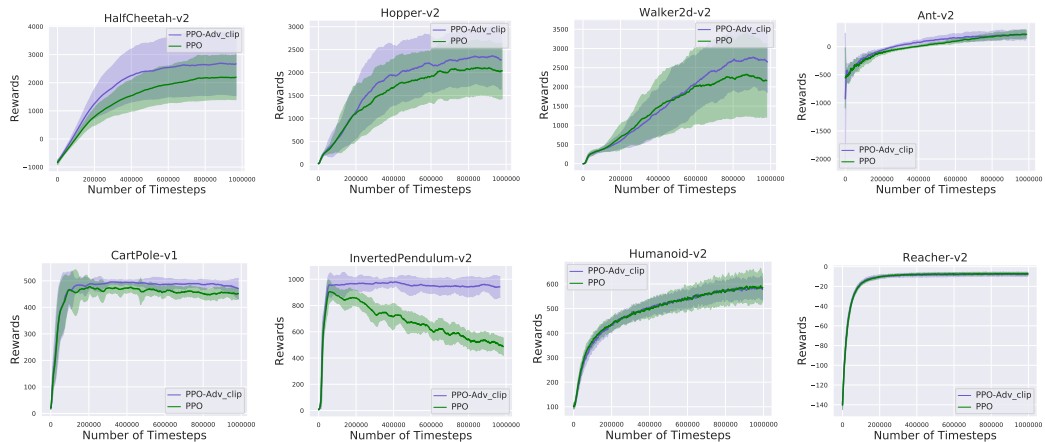

Figure 17: **Reward curves with advantage clipping in 8 different Mujoco Environments** aggregated across 30 random seeds. The shaded region denotes the one standard deviation across seeds. The clipping threshold is tuned per environment. We observe that by clipping *outlier* advantages, we substantially improve the mean rewards for 5 environments. While for the remaining three environments, we didn't observe any differences in the agent performance.

and PPO-NOCLIP. Recall that while PPO implements a clipping heuristics in the likelihood ratios (as a surrogate to approximate the KL constraint of TRPO), we remove that clipping heuristics in PPO-NOCLIP.

Engstrom et al. (2019) pointed out that trust-region enforced in PPO is heavily dependent on the method with which the clipped PPO objective is optimized, rather than on the objective itself. Corroborating their findings, we indeed observe that with optimal parameters (namely small learning rate used in our experiments), PPO-NOCLIP indeed manages to maintain a trust region with mean KL metric (Fig 16) on all 8 MuJoCo environments. This highlights that instead of the core algorithmic objective used for training, the size of the step taken determines the underlying objective landscape, and its constraints. On a related note, Ilyas et al. (2018) also highlighted that the objective landscape of PPO algorithm in the typical sample-regime in which they operate can be very different from the true reward landscape.

## J HOW HEAVY-TAILEDNESS AFFECT TRAINING?

To understand the effects of heavy-tailedness, we perform two ablation experiments: First, we study the effects of heavy-tailedness in negative advantages. By clipping the *outlier* negative advantages used in the PPO loss, we show that the induced heavy-tailedness in advantages gets reduced and the performance of the agent improves. Second, we seek to understand the effects of heavy-tailedness induced in the likelihood ratios during off-policy training. We vary the number of off-policy gradient steps (a hyperparameter otherwise fixed to 10) taken by an agent trained with PPO-NOCLIP[2] and show that the additional off-policy learning exacerbates heavy-tailedness in ratios and directly affects the agent performance even while maintaining trust-region with mean KL metric.

Next, we detail these experiments and highlight how the heavy-tailedness induced hurts the learning of agent with the mean reward metric.

### J.1 EFFECT OF HEAVY-TAILEDNESS IN ADVANTAGES

Recall that analysis in Section 3.1 shows that multiplicative advantage estimate in the PPO loss is a significant contributing factor to the observed heavy-tailedness. While PPO clipping mechanisms ameliorate the observed heavy-tailedness to some extent, the heuristics majorly offset the

---

[2]We tried using PPO loss for this experiment. However, while training with PPO, the successive policies diverge in mean KL metric when we increase the number of offline epochs used for training.

heavy-tailedness induced during off-policy gradient steps and doesn't seem to handle the induced heavy-tailedness in advantages. Motivated by this connection, we aim to study the effects of *clipping advantages* to understand its impact on the agent's behavior. In particular, we clip negative advantages which are the primary contributors to the induced heavy-tailedness.

Depending on the individual environment and the observed heavy-tailedness, we tune an environment dependent clipping threshold for advantages to maximize the performance of the agent trained with PPO. Intuitively, we expect that clipping should improve optimization and hence should lead to improved agents' performance. Corroborating the intuition, we observe significant improvements (Figure 17). Moreover, to understand the nature of clipped advantages, we plot the trend of heavy-tailedness in advantage estimates during training. Confirming our hypothesis, as we clip negative advantages below the obtained threshold, we observe that the induced heavy-tailedness stays constant throughout training (Figure 18).

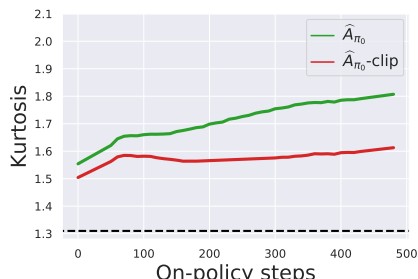

Figure 18: **Heavy-tailedness in PPO advantages with per-environment tuned clipping threshold** in MuJoCo environments. All plots show mean kurtosis aggregated over 8 Mujoco environments. With clipping advantages at appropriate thresholds (tuned per environment), we observe that the heavy-tailedness in advantages remains almost constant with training.

Our experiment unearths a previously unknown fact. Since the advantage estimates significantly contribute to the observed heavy-tailed behavior, we show that clipping outlier advantages stabilizes the training and improves agents' performance on 5 out of 8 MuJoCo tasks. While tuning a clipping threshold per environment may not be practical, the primary purpose of this study is to illustrate that heavy-tailedness in advantages can actually hurt the optimization process, and clipping advantages lead to improvements in the performance of agent. We believe that this opens a door for future research to develop more principled (and environment agnostic) methods to handle the induced heavy-tailedness in advantages which is not handled by PPO.

## J.2   EFFECT OF HEAVY-TAILEDNESS IN LIKELIHOOD-RATIOS

Recall, in Section 3.2, we demonstrated the heavy-tailed behavior of gradients during off-policy training which increases with off-policy gradient steps in PPO-NOCLIP. Moreover, we observe a corresponding increase in the heavy-tailedness of likelihood ratios. Motivated by this connection, we train agents with increased off-policy gradient steps to understand the effect of the likelihood-ratios induced heavy-tailedness on the performance of the agent. With PPO-NOCLIP, we train agents for 20 and 30 offline epochs (instead of 10 in Table 1) and analyse its performance in terms of mean reward. Note that even with 20 and 30 offline epochs the agent maintains a KL based trust-region throughout training (Bottom row in Figure 21)[3].

First, as expected, we observe an increase in heavy-tailedness in the likelihood ratios with escalated offline training (Figure 20). Moreover, the heavy-tailedness in advantages remains unaffected with an increase in the number of offline epochs (Figure 19) confirming that the observed behavior is primarily due to heightened heavy-tailedness in likelihood ratios. We hypothesize that induced heavy-tailedness can make the optimization process harder. Corroborating this hypothesis, we observe that as the number of offline epochs increases, the performance of agent trained with PPO-NOCLIP deteriorates, and the training becomes unstable (Top row in Figure 21).

The findings from this experiment clearly highlight the issues due to induced heavy-tailedness in likelihood ratios hindering efficient policy optimization. While offline training enables sample efficient training, restricting the number of off-policy epochs allows effective tackling of optimization issues induced due to the heavy-tailed nature (which are beyond just trust-region enforcement). In the future, we believe that these insights can be used to develop better algorithms for efficient offline training.

---

[3]Beyond 30 offline steps, successive policies often diverge—failing to maintain a KL based trust region.

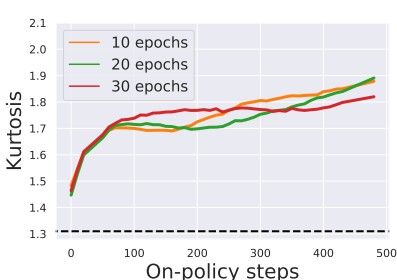

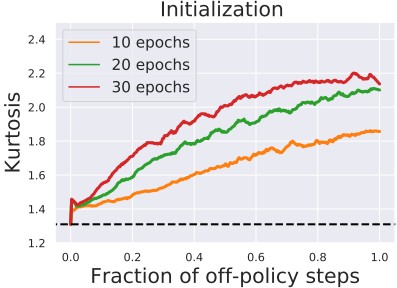

Figure 19: **Heavy-tailedness in PPO-NOCLIP advantages throughout the training as the degree of off-policyness is varied** in MuJoCo environments. Kurtosis is aggregated over 8 Mujoco environments. We plot kurtosis vs on-policy iterates. As the number of off-policy epochs increases, the heavy-tailedness in advantages remains the same showing an increase in the number of offline epochs has a minor effect on the induced heavy-tailedness in the advantage estimates.

Figure 20: **Heavy-tailedness in PPO-NOCLIP likelihood-ratios as the degree of off-policyness is varied.** at initialization in MuJoCo environments. Kurtosis is aggregated over 8 Mujoco environments. We plot kurtosis vs the fraction of off-policy steps (i.e. number of steps taken normalized by the total number of gradients steps in one epoch). As the number of off-policy epochs increase, the heavy-tailedness in ratios increases substantially. The same trend holds at other training iterations, however, the degree of increase tappers off.

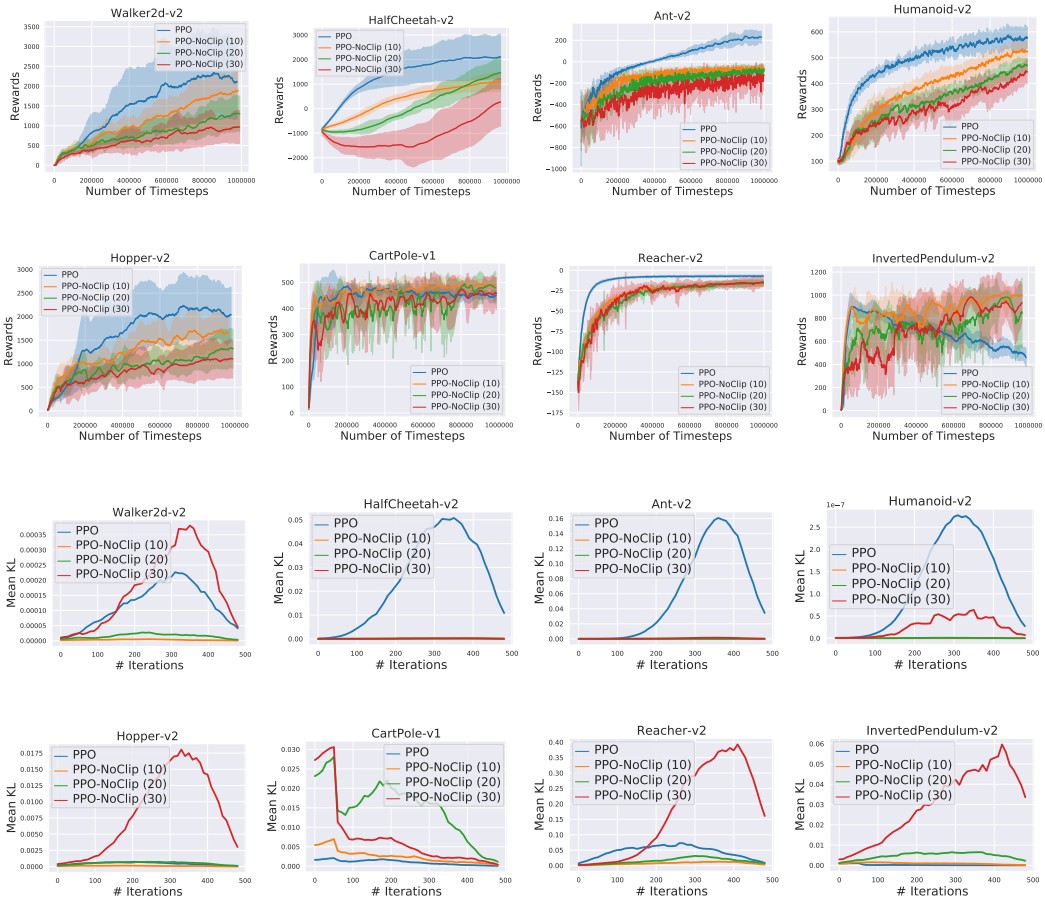

Figure 21: **(Top two rows)** Reward curves with the varying number of offline epochs in 8 different Mujoco Environments aggregated across 10 random seeds. Bracketed quantity in the legend denotes the number of offline epochs used for PPO-NoClip training. Clearly, as the number of offline epochs increases, the performance of the agent drops (consistent behavior across all environments). Furthermore, at 30 epochs the training also gets unstable. We also show the PPO performance curve for comparison. **(Bottom two rows)** KL divergence between current and previous policies with the optimal hyperparameters (parameters in Table 1) for PPO and PPO-NoClip with varying number of offline epochs. We measure mean empirical KL divergence between the policy obtained at the end of off-policy training and the sampling policy at the beginning of every training iteration. The quantities are measured over the state-action pairs collected in the training step. We observe that till 30 offline epochs PPO-NoClip maintains a trust-region with mean KL metric.

