# OpenReview forum: "On Proximal Policy Optimization's Heavy-Tailed Gradients "
_ICLR.cc/2021/Conference — Reject_

### Official Review · AnonReviewer2 · 2020-10-23
**Important advance into understanding of PPO, with some caveats about statistical significance**

**Rating:** 8
**Confidence:** 4

**Review:**

**Positives**

The claims of the paper are clear and the evidence to support the claims is clearly presented.

The question of heavy-tailedness is well-motivated and important. There has been growing interest in characterizing the tails of gradient norms, especially in light of assumptions in optimization theory that may be invalidated by heavy tails (e.g., second moment bound). Characterization of the heavy tail property permits further understanding of why our algorithms might fail, and how we might improve them.

The central claims are well-substantiated with experiments (subject to statistical significance concerns below). Sections 3.1-3.2 show that heavy-tailed gradient norms occur both in the on-policy and off-policy parts of PPO. I especially liked the care in determining the source of the heavy-tailedness: the advantage in the on-policy case and the IS ratio in the off-policy case. Section 3.3 shows that the clipping of the PPO objective reduces the heavy-tailedness of the gradients, which is further support for the relevance of heavy-tailedness.

I appreciated the promising connection made to the field of robust statistics. In section 4, the paper modifies a robust gradient estimation technique to apply in PPO. The fact that Robust-PPO-NoClip does either as well (again, subject to statistical significance concerns below) as or better than PPO or its NoClip variant is evidence that not accounting for heavy-tailed gradients can impede performance.

**Negatives**

The main problem with this work is the relative lack of statistical significance analysis (figs 1-4). I would have liked to see error bars on figs 1-4. For the individual learning curves (fig 11) in the appendix, it seems that more runs are necessary given the noisiness of the curves. Are there more disaggregated plots of the kurtoses over the environments (in addition to the three in the appendix)? I'm not totally sure what to make of averaging over environments whose dynamics could be quite different. For each environment, it also seems that only 10 seeds were run. Given that prior work (Henderson et al., 2017) suggests that RL algorithms are extremely sensitive to the random seed, I would have liked to see more seeds (minimum 20, preferably 30+).

The work needs some further discussion on why Robust-PPO-NoClip only seemed to beat PPO-NoClip in some environments, but not others.

**Summary**

I'm leaning towards a marginal accept for the paper: I do think the contribution of the paper is valuable, despite the concerns I have outlined above about statistical significance. I think these concerns are strongest for the control experiments (a relatively smaller part of the paper) given that they are per-environment and utilize fewer samples to estimate the return, as opposed to the gradient estimation experiments for which a batch size of 64 was used for the gradient norm estimation. I think this paper otherwise provides much value in characterizing an important property of PPO, opening the way for future RL algorithms that can better deal with heavy-tailedness.

**Things which didn't affect my decision**

1. Uniform axes for fig 7 would be helpful, to permit comparison of plots


** Edit after author response **
The authors have addressed the concerns I had so I've increased the score accordingly.

---

> ### Author Response · Authors · 2020-11-19
> **Author Response**
>
> We thank the reviewer for the constructive review and positive feedback. We are also glad that you consider the contributions of our work as valuable and opening the way for future RL algorithms that can better deal with heavy-tailedness.
>
> **“For each environment, it also seems that only 10 seeds were run … I would have liked to see more seeds (minimum 20, preferably 30+).”**  Inspired by your suggestions, we have updated the control and Kurtosis analysis experiments with 30 seeds. The corresponding plots have been updated in the paper. We note that all our empirical observations remain the same with more seeds.
>
> **“Are there more disaggregated plots of the kurtoses over the environments (in addition to the three in the appendix)?”** We have updated the Figures (12-14) in Section H in the paper to include curves on all 8 individual environments.
>
> **“… the relative lack of statistical significance analysis (figs 1-4). I would have liked to see error bars on figs 1-4. I'm not totally sure what to make of averaging over environments whose dynamics could be quite different.”** We aggregated the results from individual environments to show the trend observed (which is common to all environments). As per your suggestion for completeness, we have updated the draft with plots on individual environments with the same analysis and error bars. We omitted error bars in Figures 1-4 because the magnitude of kurtosis value varies a lot across different environments confusing the standard deviation values. We have clarified this in the main paper.
>
> **“The work needs some further discussion on why Robust-PPO-NoClip only seemed to beat PPO-NoClip in some environments, but not others.”** We observe that Robust-PPO-NoClip beats PPO-NoClip in all but Hopper-v2. While we don’t fully understand the underlying cause for such behavior, we suspect that this might be an artifact of the bias in the gradient estimate of Robust-PPO-NoClip.
>
> **“Uniform axes for fig 7 would be helpful, to permit comparison of plots.”** We have updated the plot accordingly.

---

> > ### Comment · AnonReviewer2 · 2020-11-21
> > **author response**
> >
> > Thank you for the response! I will increase my score accordingly. I really appreciate you running the additional experiments.

---

### Official Review · AnonReviewer4 · 2020-10-27
**Interesting empirical analysis**

**Rating:** 7
**Confidence:** 4

**Review:**

**Update Post Rebuttal:** The revised manuscript addresses my main concerns, and is stronger than the original, so I am raising my score to accept. For the final manuscript, I urge the authors to better integrate the added content from this discussion period into the main text, instead of relegating it to the Appendix.

**Summary**: This paper hypothesizes that the policy gradient in RL has heavy tails, and conducts a thorough empirical investigation of the heavy-tailed properties of PPO and it's critic estimation. The paper finds that advantages and the sampling ratio have the heaviest tails. Based on this insight, the authors propose a robust version of the PPO gradient estimator that avoids clipping and heuristics, but which still generally matches performance with PPO.

**Assessment:** The paper tests a simple, important, and previously un-studied hypothesis, and provides a convincing empirical analysis. It provides a potential step towards designing better RL algorithms without the heuristics and sensitivity of current RL algorithms, and as such, would be of great interest to the reinforcement learning community at ICLR. Nevertheless, there are some major concerns detailed below, primarily relating to the validity of the PPO-NoClip objective. Conditional on these concerns being addressed, I would recommend acceptance.

**Major Comments:**

While the likelihood clipping term in PPO is certainly heuristic, it serves a purpose that is not properly discussed in the paper: to prevent $\pi_\theta$ from deviating from $\pi_0$. More precisely, when function approximation is used for the policy, the surrogate objective $E_{s \sim \pi_0, a\sim \pi_\theta}[A_{\pi_0}(s, a)]$ is not necessarily a good surrogate when $\pi_\theta$ is far from $\pi_0$ **even with infinite samples**. This is not an issue with finite sampling as is mentioned in the preliminaries, and remains a problem even with access to the true expectation (that is, even when heavy-tailed behavior is not a problem). See the TRPO paper [1] for more discussion. This point needs to be made more explicit in the main paper.

As a result, it is my understanding PPO-NoClip is simply not an appropriate gradient / objective to be analyzing when $\pi_\theta$ is very far from $\pi_0$. This is a problem for the off-policy analysis, since the PPO-NoClip objective and its gradient do not relate to the original expected return objective, and so may not be meaningful quantities to measure or analyze. My initial assessment is that this highly reduces the value of the off-policy experiments for me.

**Clarity**: The paper is generally well-written and easy to follow. However, the paper could greatly benefit from a discussion why heavy-tailed gradients are bad in reinforcement learning. The paper does a very good job of arguing that heavy-tailed gradients exist in PPO, but has little discussion or empirical evidence to suggest that this is necessarily a problem.


**Minor Comments:**

- Some recent work [2] also studies a similar phenomenon more generally in stochastic PG methods, and find that second-order moments are insufficient to describe learning phenomena, and would be worth citing.
- It would be useful to write the exact equation / objective used for `PPO-NoClip` in the main paper
- In Section 2.1, it is only stated that Pareto distributions (with lower $\alpha$) also have higher kurtosis. Does some statement to this effect hold for arbitrary distributions (not Pareto)?
- Why are results different for PPO and A2C in the On-Policy iterations? If only on-policy iterations are conducted, aren't the PPO and A2C updates the same (both equivalent to standard policy gradients).
- I found the x labels for Figure 1 and Figure 2 confusing. From what I understood, Figure 1 is taken across iterations (different data collection policies for each step) but Figure 2 is taken within a single iteration? This could be more clear.
- I found the obervation that the negative advantages are the ones with heavy tails to be interesting. However, given the fact that the PPO objective is to minimize the probability of highly negative advantages, aren't heavy tails to be expected? The discussion also didn't make it clear to me why having heavy tails for negative advantages is necessarily a bad thing.
- In the Preliminaries, the policy in general doesn't map to [0,1] but in the continuous setting, more generally to $\mathbb{R}_+$.

[1] Schulman et al. Trust Region Policy Optimization, ICML 2015

[2] Chung et al. Beyond variance reduction: Understanding the true impact of baselines on policy optimization. 2020. ArXiv

---

> ### Author Response · Authors · 2020-11-19
> **Author Response (Part 1)**
>
> Thank you for the detailed and thoughtful review and a positive assessment of our paper. We are glad to see that you consider our work as a potential step towards designing better RL algorithms without the heuristics and sensitivity of current RL algorithms.
>
> **"… the likelihood clipping term in PPO is certainly heuristic, it serves a purpose that is not properly discussed in the paper: to prevent \\(\pi_\theta\\)  from deviating from \\(\pi_0\\)."** Thanks for pointing this out. We completely agree with the comment and as per your suggestion, we have made the appropriate corrections and added a discussion explaining the purpose of the KL constraint used in TRPO and clipping heuristic in PPO in Preliminaries (Section 2).
>
> **"... PPO-NoClip is simply not an appropriate gradient/objective to be analyzing when \\(\pi_\theta\\) is very far from \\(\pi_0\\)"** Thank you for the insightful detailed comment discussing the validity of PPO-NoClip experiments. Motivated by the recent work (Engstrom et al. 2019), we analyze successive policies (sampling policy, i.e., the policy used for sampling a batch of data and the policy we obtain after 320 gradient steps, i.e., at the end of training iteration) to understand the trust-region maintained with mean KL divergence. To justify the use of PPO-NoClip in our analysis experiments, we show that with optimal hyperparameters used in experiments, agents trained with PPO-NoClip maintains a KL based trust-region like PPO throughout the training.  We have added a new section in the Appendix (Section I, Figure 16) elaborating this.
>
> **"… but has little discussion or empirical evidence to suggest that this is necessarily a problem. ... didn't make it clear to me why having heavy tails for negative advantages is necessarily a bad thing?"** We have updated the draft with additional discussion and ablation experiments supporting the hypothesis that heavy-tailedness hinders efficient optimization (Section J) and showing that by addressing it, we can achieve stable learning and superior performance. We performed two experiments:
>
> (i) First, we study the effects of heavy-tailedness in negative advantages. By clipping the outlier negative advantages (with an environment dependent threshold) in the PPO loss, we show that the induced heavy-tailedness in advantages gets reduced and the performance of the agent improves. Since the advantage estimates significantly contribute to the observed heavy-tailed behavior, we show that clipping outlier advantages stabilizes the training and improves agents' performance on 5 out of 8 MuJoCo tasks. While this may not be practical (due to per environment thresholds), the primary purpose of this study is to illustrate that heavy-tailedness in advantages actually hurt the optimization process, and clipping advantages lead to improvements in the agents' performance (Section J.1).
>
> (ii) Second, we seek to understand the effects of heavy-tailedness induced due to heavy-tailedness in the likelihood ratios induced during off-policy training. We vary the number of off-policy gradient steps taken by an agent trained with PPO-NoClip and show that the additional off-policy learning exacerbates heavy-tailedness in ratios and directly affects the agent performance even while maintaining trust-region with mean KL metric. The findings from this experiment clearly highlight the issues due to induced heavy-tailedness in likelihood hindering efficient policy optimization (Section J.2).
>
>
> **“Some recent work [2] also studies a similar phenomenon more generally in stochastic PG methods ... would be worth citing”** Thanks for pointing this out. We have included a discussion of this in our related work section.
>
> **“In Section 2.1, it is only stated that Pareto distributions (with lower alpha) … effect hold for arbitrary distributions (not Pareto)?”** In general, the distributions which have Kurtosis greater than that of the normal distribution are called Leptokurtic. In terms of shape, a leptokurtic distribution has fatter tails. Examples of leptokurtic distributions include the Student's t-distribution, Rayleigh distribution, Laplace distribution, exponential distribution, Poisson distribution, and the logistic distribution.

---

> > ### Author Response · Authors · 2020-11-19
> > **Author Response (Part 2)**
> >
> > **“Why are results different for PPO and A2C in the On-Policy iterations? .. aren't the PPO and A2C updates the same”** We apologize for any confusion here. When we consider on-policy iterations with PPO, we do not alter the training of the agent with PPO. We just consider the first gradient step made on a new batch of sampled data such that optimizing policy is the same as the policy from which the data was sampled. But the agent (when trained with PPO) is still trained to optimize the objective with off-policy data. Whereas with A2C, there is no off-policy training -- every gradient step is taken on a new batch of data.
> >
> > Since with PPO, the agent is trained for extra gradient steps which particularly are taken on the off-policy data, we observe that the off-policy training of the critic network contributes to the heightened heavy-tailedness in advantages and hence in the on-policy gradients. We have added a plot showing the trend of advantages for PPO and A2C in Appendix E.2.
> >
> > **“Figure 1 and Figure 2 confusing... Figure 1 is taken across … different data collection policies for each step but Figure 2 is taken within a single iteration?”** Yes, this is correct. We have improved the exposition to make this clearer in the paper.
> >
> > **“However, given the fact that the PPO objective is to minimize the probability of highly negative advantages, aren't heavy tails to be expected?”** We agree that minimizing the objective in PPO can lead to negative advantages. However, we do not fully understand the exact underlying causes as clearly as in Bandit settings (Shin et. al. 2019). We believe that understanding the exact causes of outliers in negative advantages for continuous control tasks can be an interesting problem for future studies.
> >
> > **“It would be useful to write the exact equation/objective used for PPO-NoClip in the main paper”** Done, included in Appendix A.
> >
> > We have also fixed other typos and notation issues pointed out.

---

> > > ### Comment · AnonReviewer4 · 2020-11-23
> > > **Satisfied with the Author Response**
> > >
> > > Thank you for the response. In my opinion, the revised manuscript addresses my main concerns, and is stronger than the original, so I am raising my score.
> > >
> > > For the final manuscript, I urge the authors to better integrate the added content from this discussion period into the main text, instead of relegating it to the Appendix. In particular, I'd recommend moving  the discussion on why heavy-tailed gradients negatively affect from Appendix J to the main text.
> > >
> > > Figure 16 caption: peeks -> peaks

---

### Official Review · AnonReviewer3 · 2020-10-27
**A interesting empirical analysis of some RL gradient statistics, but unclear takeaways**

**Rating:** 5
**Confidence:** 3

**Review:**

### Strengths

This paper presents an intriguing analysis of the gradient distributions over the course of training for popular RL algorithms in common mujoco benchmarks.  The observation that negative advantages are a bigger contributor to the kurtosis than positive advantages seems interesting and if true as a general phenomenon, deserving of more understanding. The authors also propose a new alternative (inspired by robust statistic) to the simple PPO clipping heuristic that does reasonably well even if it doesn't deliver any clear improvements over PPO.

### Weaknesses

The paper is well motivated, and has a collection of interesting observations but I am not sure if these empirical observations lead to something beyond that, in terms of a more general claim/conjecture. The algorithmic contribution, while novel, is neither particularly strong in terms of performance nor simpler compared to the PPO clipping heuristic.

### Comments

* By the heavy tailedness in "advantage divided gradients", can the authors refer to exactly what 1-dimensional projection is being referred to? This is something ambiguous in the legend in Figure 1 (b) and (c) as well.

* Figure 6 in Appendix E is used to support the claim that the kurtosis increases over time only for negative advantages, but in the figure this isn't clear to me. e..g. for returns there seems to be an increase over time. Also, nit: typo in the legend for $\hat{V}$.

* "This observation highlights that at least in Mujoco ...., there is a positive bias of ....value estimate....for actions with negative returns" <-- this seems a little ambiguous, are you referring to negative advantages?

* In the off-policy case, Figure 2 suggests that the heavy tailedness is coming only from the importance weight ratios. What happened to the heavy tailedness of the (negative)advantages being another source? It seems a bit mysterious that this is only present in the on-policy case and disappears when off-policy. In particular why would the heavy tailedness in the importance ratios and the heavy tailedness from advantages be mutually exclusive?

* The setup for Sec 3.2 is not clear even with appendix D. What does the iteration in the learning curve refer to when considering a particular snapshot like 50% of the max reward?

* Nit/typo: The notation for the GMOM description in the text is missing the \hat for $\mu_b$.

---

> ### Author Response · Authors · 2020-11-19
> **Author Response**
>
> We thank the reviewer for their thoughtful feedback. We are glad that you find our analysis intriguing.
>
> **"… but I am not sure if these empirical observations lead to something beyond that ... The algorithmic contribution, while novel, is neither particularly strong in terms of performance nor simpler compared to the PPO"** While we do not obtain improvements over PPO, the main focus of our work is to understand the nature of gradients (and contributions of various loss components in that behavior) in importance-weighted policy gradients and provide a fresh perspective to PPO heuristics that are poorly understood. We highlight the heavy-tailed nature of gradients and discern the underlying factors. Now, we have also added empirical results (in Appendix J) demonstrating the heavy-tailed nature of gradients degrades the performance of the agent and causes instability.
>
> The main takeaway from our robust gradient estimation is to highlight that even without PPO clipping heuristics we can achieve close to PPO performance with a method that is principally designed to fix the heavy-tailedness issue. We believe that this strengthens our conjecture of heavy-tailedness being an issue with optimization.
>
>
> **"By the heavy tailedness in advantage divided gradients, what 1-dimensional projection is being referred to? "** Due to the nature of the loss function, the same advantage estimate (scalar) gets multiplied with each coordinate of the gradient.  Since the advantages scale each coordinate of the gradient with the same value (magnitude and scale), we just run the analysis with the gradient vector obtained by dividing the original gradient coordinatewise with the corresponding advantage estimate. Throughout the paper, we have analyzed three different estimators. For Kurtosis, we perform analysis on gradient norms.
>
> **"... Figure 2 suggests that the heavy tailedness is coming only from the importance ratios. What happened to (negative) advantages being another source? a bit mysterious that this is only present in the on-policy case and disappears when off-policy."** We apologize for the confusion here. Yes, the heavy-tailedness in the ratios and advantages are not mutually exclusive -- i.e. heavy-tailedness in advantages doesn’t disappear when off-policy. In fact, as mentioned in the caption of Figure 2, the heavy-tailed nature of advantages is the main cause of an overall increase in heavy-tailedness across training iterations.
>
> The main takeaway that Figure 2 highlights is that we demonstrate an increase in heavy-tailedness during off-policy training, which is caused by likelihood ratios. Note that since the advantage estimates are not updated during off-policy steps, the heavy-tailedness in advantages remains constant while taking off-policy gradient steps in any specific training iteration.
>
>
> **"Figure 6 in Appendix E is used to support the claim that the kurtosis increases over time only for negative advantages, but in the figure this isn't clear to me. e..g. for returns there seems to be an increase over time"** Thanks for pointing this out. We have updated the caption and text in the main paper to capture the slight increase in the heavy-tailed nature of the returns. However, the minor heavy-tailedness in returns (which plateaus quickly as training progresses) does not explain the increasing nature of heavy-tailedness in negative advantages.
>
> **"… this seems a little ambiguous, are you referring to negative advantages?"** Thanks for catching. Fixed.
>
> **"The setup for Sec 3.2 is not clear even with appendix D. What does the iteration in the learning curve refer to with snapshot like 50% of the max reward?"** We have improved the exposition in the appendix to elaborate on this. We elaborate the exact setup at 50% of the maximum reward. First, we find the training iteration where the agent achieves approximately 50% of the maximum reward individually for each environment. Then at this training iteration, we freeze the policy and value network and save the sample-wise gradients of the actor and critic objective for off-policy steps.
>
> We have also fixed other typos and notation issues pointed out. Thanks for catching these errata.

---

### Official Review · AnonReviewer1 · 2020-10-30
**Interesting analysis but needs more work**

**Rating:** 5
**Confidence:** 4

**Review:**

**Summary**: This paper performs an empirical analysis of the heavy-tailedness of the PPO gradients across MuJoCo environments. They find that PPO gradients are heavy-tailed, which means that they are sensitive to outliers, which means computing the expected gradient is hard. The paper studies two causes of this issue -- advantage estimation errors and the harm caused by optimizing density ratios in a sampled setting, and shows that removing either of them with the other issue controlled for helps PPO. They then propose to use a standard robust mean estimation technique to obtain a robust gradient mean estimator that is plugged into PPO, and performs sort of somewhat worse than vanilla PPO.

I like the paper, but I do think the paper misses some comparison to prior work and alternate views of looking at the same phenomenon which I will discuss next. I am not sure if heavy-tailedess of gradients is the best way of looking at it. Very similar phenomena have been studied in the form of using regularization to prevent overfitting to the density ratios in off-policy bandits (see Swaminathan et al. 2015), and also the impact of variance reduction techniques such as the use of baselines in policy optimization (several papers). What these papers show is that (1) optimizing against importance weighting in the presence of samples may not lead to an improvement in the overall objective, which is bad -- and some on-policy regularization is employed (2) baselines help because they make it possible to reduce variance, however, their form also impacts learning dynamics and exploration (see Chung et al., Beyond variance reduction: Understanding the true impact of baselines on policy optimization) and just looking at variance may not be enough to talk about the efficacy of a baseline. So baselines affect optimization issues even in an expected sense, and just variance is not the best way of looking at the efficacy of a baseline. Translated to PPO, this means that some advantage estimators that lead to higher variance gradients or heavy-tailed gradients might actually be better because the learning dynamics and exploration is better here. Therefore, these are alternate ways of explaining the same issue that the paper points out. So, why is heavy-tailedness interesting, and why should we not look at these other interpretations?

In particular, is heavy-tailedness always bad? I suspect not -- in some cases, I might want to have more heavy-tailed gradients just because they can either help to navigate the optimization landscape (see Mei et al. On the Global Convergence of Softmax Policy Gradient Methods, or Schaul et al. Ray Interference: a Source of Plateaus in Deep Reinforcement Learning or Chen et al. Surrogate Objectives for Batch Policy Optimization in One-step Decision Making, for a discussion of how the true expected policy gradient can have many saddle points and can have slow convergence based on initialization) by avoiding getting stuck at saddle points. So maybe it is not always that bad? Perhaps some analysis in settings where optimization challenges exist and a comparison of the impact of heavy-tailedness in scenarios where optimization challenges don't exist is a possible way to answer this question.

Also what happens if I compute the true advantages? Heavy tailedness based on your presented analysis would decrease but would it be better to still do robust gradient estimation? I suspect there will still be some sort of overfitting due to finite sample reuse in the off-policy updates in PPO, but it is interesting to see how much the performance improves in that case. And what would happen in methods that actually derive policy updates from a closed-form objective such as REPS, RWR/AWR, SIL, MPO, etc?

Finally, the method seems like fixing an effect and not the cause -- if the end goal is to obtain better advantage estimates and do better optimization with density ratios, why not fix that directly, rather than using robust mean estimation, which seems like a complex procedure? One way of fixing advantages, for instance, is to just utilize a monotonic function of advantages such as the indicator function in Self-imitation learning and exponentiation function in AWR. Or for instance, centering the advantages which PPO uses can also help (see https://openreview.net/pdf?id=SJaP_-xAb) and maybe even using self-normalization on the importance weights which are optimized in PPO. How would these solutions compare to the proposed estimator?

I think that the paper will benefit from a more concrete discussion of these aspects, especially contrasting and clearly stating the benefits of the proposed view of understanding PPO in comparison with prior explanations and by doing a more extensive evaluation of simpler alternatives to fixing this issue.

---

> ### Author Response · Authors · 2020-11-19
> **Author Response**
>
> We thank the reviewer for the thoughtful and detailed review.
>
> **"... but I do think the paper misses some comparison to alternate views ... I am not sure if heavy-tailedess is the best way of looking at it."** Thanks for pointing out relevant references. We are working to add a discussion of these papers in our related work section. We do not assert categorically that heavy-tailedness of gradients is the only useful for understanding \*\*all\*\* optimization issues in importance weighted policy gradients. However, we do think that this a promising perspective on the problem, as demonstrated by our analysis here unearthing some previously unknown issues and providing a more principled framework for analyzing and fixing these issues.
>
> Our analysis provides a novel perspective to PPO heuristics that are poorly understood (Ilyas et al., 2018, Engstrom et al., 2019). Inspired by Engstrom et al., we show that PPO-NoClip also maintains a KL based trust-region when trained with tuned hyperparameters (App I).  We argue that in practice, PPO clipping heuristics are tackling issues beyond just maintaining a trust-region. Moreover, we demonstrate that even while PPO doesn’t maintain a ratio based trust-region despite being directly trained with ratio-clipping objective (Engstrom et al.), the clipping trick serves to alleviate heavy-tailedness which significantly improves the performance of the agent.
>
> To summarize, we are not claiming that our perspective provides a full account of the dynamics of PPO (and indeed other interpretations can provide useful explanations to some problems). We just believe that this framework provides a novel perspective to instability issues in deep RL, enjoys some experimental support for the usefulness of the perspective, and can potentially open the door to more principled algorithmic developments in deep RL.
>
> **"Is heavy-tailedness always bad?" and comparison with Chung et. al.** Thanks for pointing this out. It’s interesting to note that larger higher moments with fixed variance can lead to improved exploration in simple MDPs. This is also related to one view of heavy-tailedness in supervised learning where authors conjectured that heavy-tailedness in gradients can improve generalization (Simsekli et. al. 2019).
>
> However, there is another view to heavy-tailedness where outliers cause instability in the learning process in deep models (Zhang et. al. 2019a;b). We conjecture that in deep RL where the optimization process is known to be brittle (Henderson et. al. 2017;2018), heavy-tailedness can exacerbate instability and this is a more prominent effect than any help it might offer in efficient exploration. Indeed with ablation experiments (added in App J), we show that increasing heavy-tailedness in likelihood ratios hurt the agent’s performance, and mitigating heavy-tailedness in advantage estimates improves learning dynamics and hence the agent’s performance.
>
> **"What happens if I compute the true advantages? Heavy tailedness would decrease but better to still do robust estimation?"**  Our analysis shows two primary contributing factors to the heavy-tailed behavior: One, due to induced heavy-tailedness in likelihood-ratios and the other in advantages estimates. We added a new section (App J) where we demonstrate how heavy-tailedness in ratios and advantage estimates individually contribute to the optimization issues in PPO.
>
> While fixing the heavy-tailedness in advantages will remove one source, the heavy-tailedness induced during off-policy training would still be an issue. Moreover, as shown in Sec 3.3, PPO clipping heuristics primarily serve to offset the heavy-tailedness induced during off-policy gradient steps---highlighting the need to tackle heavy-tailedness in ratios. Note that PPO-NoClip that doesn’t handle the ratios induced heavy-tailedness and performs much worse than PPO.
>
> We also have a clarification question regarding the suggested experiment. We are not sure about how to compute true advantages in MuJoCo environments with PPO and we request if the reviewer can further clarify this.
>
> **"Finally, the method seems like fixing an effect and not the cause …"** Yes, we agree that we fix the effect and don’t alter the underlying cause. But the main contribution of our work is not developing RL algorithms that improve performance over PPO, but rather to understand the nature of gradients (and contributions of various loss components in that behavior) in importance-weighted policy gradients.
>
> The main takeaway from our robust estimation is to highlight that even without PPO clipping heuristics, we can achieve close to PPO performance with a method that is principally designed to fix the heavy-tailedness issue. We believe that this strengthens our conjecture of heavy-tailedness being an issue with optimization. Moreover, as discussed above, we added an ablation experiment that provides more support for the hypothesis that heavy-tailedness hinders efficient optimization.

---

### Author Response · Authors · 2020-11-19
**Updated draft with new experiments and clarifications**

We thank all four reviewers for their detailed and thoughtful feedback. Overall, the reviewers found the analysis intriguing and appreciated the connections made with robust statistics. We are grateful to all the reviewers for numerous constructive suggestions. Inspired by this feedback, we have run several experiments and added them to the draft. We summarize the key changes below:

**1. Updated experiments with 30 seeds:** As per R2’s suggestion, we have updated plots by aggregating over 30 seeds to address statistical significance concerns (we note that our experimental observations remain the same with more seeds).

**2. KL divergence analysis:** We have also included KL divergence analysis with PPO-NoClip where we find that the successive policies (during off-policy training) do maintain a KL based trust-region that addresses R4’s concerns about the validity of PPO-NoClip experiments.

**3. Issues arising due to Heavy-tailedness in PPO gradients:** We added two ablation experiments to test how heavy-tailedness in likelihood-ratios and advantage estimates individually contribute to the optimization issues in PPO. Empirically, we find that heavy-tailedness in likelihood ratios induced during off-policy gradients can be a significant factor causing optimization instability leading to low average rewards. Moreover, we show that removing heavy-tailedness in advantages can also help in improving the optimization by allowing agents to achieve superior performance. We have added a detailed discussion on this in Appendix J. (R1, R3, R4)

We have updated the draft with these experiments and some other minor experiments to address the concerns of reviewers. Changes in the revised paper are highlighted in blue.

We also want to make clarifications about our contributions. The main aim of the paper is to understand the nature of gradients (and contributions of various loss components in that behavior) in importance-weighted policy gradients and provide a novel perspective to PPO heuristics that are poorly understood. We highlight the heavy-tailed nature of gradients and discern the underlying factors. In the current draft, we have also added results (in Appendix J) demonstrating that the heavy-tailed nature of gradients degrades the performance of the agent and causes instability.

Secondly, inspired by our heavy-tailed analysis, we leverage a robust gradient estimation technique and show that it can achieve close to PPO performance without clipping heuristics. Since our primary aim is not to achieve performance improvements over PPO, we want to emphasize that the robust gradient estimation achieving close to PPO performance strengthens our conjecture of heavy-tailedness being an issue and PPO clipping heuristics addressing that issue.

Overall, we believe that developing a deeper understanding of the state of the art algorithms can potentially lead to principled algorithmic development in RL and is equally important as introducing new methods with superior performance.

Below, we respond to each reviewer in more detail.

---

### Decision · Program_Chairs · 2021-01-07
**Final Decision**

**Decision:**

Reject

**Comment:**

Dear authors,

As you can see, reviewers agree on the importance of the analysis present in the paper but two reviewers feel like it misses important comparisons.

That said, PPO is a popular algorithm and I also welcome any attempt as improving our understanding of its dynamics. With this paper, information about PPO would be more complete but also more spread out across multiple papers.

At the same time, I am sympathetic to the reviewers' arguments and also feel that the paper would have had a much clearer message had some additional ablation studies been performed, for instance on tabular settings where this is easily done.

The overall assessment is that the paper is not yet ready for publication.